# MMCSBench: A Fine-Grained Benchmark for Large Vision-Language Models in Camouflage Scenes

**Jin Zhang**    **Ruiheng Zhang**[*]    **Zhe Cao**    **Kaizheng Chen**
Beijing Institute of Technology, Beijing, China
jinzhang@bit.edu.cn
https://github.com/zhangjinCV/MMCSBench

## Abstract

Current camouflaged object detection methods predominantly follow discriminative segmentation paradigms and heavily rely on predefined categories present in the training data, limiting their generalization to unseen or emerging camouflage objects. This limitation is further compounded by the labor-intensive and time-consuming nature of collecting camouflage imagery. Although Large Vision-Language Models (LVLMs) show potential to improve such issues with their powerful generative capabilities, their understanding of camouflage scenes is still insufficient. To bridge this gap, we introduce MMCSBench, the first comprehensive multimodal benchmark designed to evaluate and advance LVLM capabilities in camouflage scenes. MMCSBench comprises 22,537 images and 76,843 corresponding image-text pairs across five fine-grained camouflage tasks. Additionally, we propose a new task, Camouflage Efficacy Assessment (CEA), aimed at quantitatively evaluating the camouflage effectiveness of objects in images and enabling automated collection of camouflage images from large-scale databases. Extensive experiments on 26 LVLMs reveal significant shortcomings in models' ability to perceive and interpret camouflage scenes. These findings highlight the fundamental differences between natural and camouflaged visual inputs, offering insights for future research in advancing LVLM capabilities within this challenging domain.

## 1 Introduction

Fine-grained cognitive understanding in automated visual perception stands as a core challenge toward achieving Artificial General Intelligence (AGI) [22, 69]. Previous challenging AI research in automated visual perception, exemplified by Camouflaged Object Detection (COD) [68, 12, 46, 36], primarily relies on discriminative models under conventional deep learning paradigms. These methods typically treat COD as a sample segmentation task, successfully identifying '*what*' to segment but failing to grasp the contextual relationships that explain '*why*' an object is camouflaged within its scene. This superficial understanding severely restricts their real-world applicability and generalization to new categories and scenarios [54]. Furthermore, systematically identifying and collecting camouflage images from large-scale natural datasets highly relies on laborious manual selection, hindering this field's development due to data scarcity. Currently, the largest benchmark dataset, COD10K [12], contains only 5,786 images with camouflaged objects for training and testing.

Recent advances in Large Vision-Language Models (LVLMs) [16, 62] present promising new avenues for tackling these challenges. Their inherent generative flexibility and ability to process multimodal inputs allow for nuanced, human-like guidance via language instructions. Notably, their demonstrated strength in understanding complex visual relationships [70, 71, 45, 18, 21] suggests they could potentially grasp the crucial scene-object context (the '*why*') missed by traditional COD approaches.

---

[*]Corresponding author. Email: ruiheng.zhang@bit.edu.cn

39th Conference on Neural Information Processing Systems (NeurIPS 2025) Track on Datasets and Benchmarks.

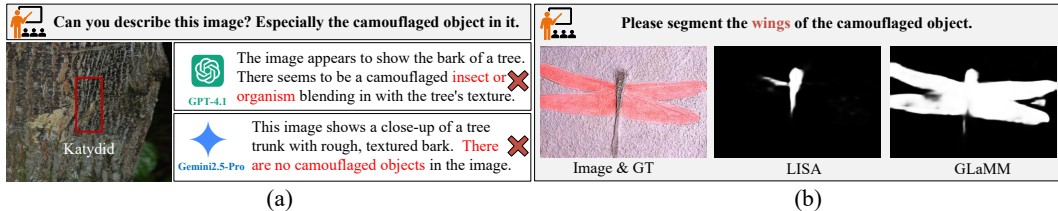

Figure 1: **Performance of existing LVLMs in understanding camouflage scenes and grounding segmentation**. Figure (a) demonstrates the camouflaged object recognition capabilities of GPT-4.1 [1] and Gemini2.5-Pro [55]. Figure (b) presents the instruction-following abilities of existing grounding segmentation models, LISA [24] and GlaMM [49], in relation to camouflaged objects.

However, directly applying existing LVLMs to camouflage scenes remains significantly challenging. Existing LVLM benchmarks [29, 37] primarily target general natural scenes containing clear and salient objects, lacking sufficient adaptability to highly camouflaged scenes. As illustrated in Fig. 1, when specifically applied to camouflage scenes, the relevant data scarcity is likely to lead to increased model hallucination and degraded instruction-following performance. Therefore, establishing a dedicated benchmark for camouflage scenes is imperative, not only to systematically evaluate LVLMs' true capabilities in these challenging scenarios but also to leverage generative LVLMs for fundamentally understanding camouflage mechanisms.

Current LVLMs face severe challenges when evaluating camouflage scenes, as they lack sufficient learning data for detecting such scenes. To address this critical gap, we introduce MMCSBench, the first-ever multimodal camouflage scene benchmark specifically designed for LVLMs. We first collect a large number of camouflage images, and then manually and meticulously annotate high-quality camouflage attribute descriptions for them. These annotations are structured based on seven key dimensions: `quantity`, `category`, `color`, `size`, `position`, `camouflage strategy`, and `camouflage target`. Using these captions along with GPT-4V [1], we generate fine-grained visual and multiple-choice questions, which five experts then answered to obtain the corresponding ground truth text. Ultimately, this process resulted in the curation of 22,537 images from five public datasets, constructing 76,843 image-text pairs across five fine-grained tasks for camouflage scene understanding, as shown in Fig. 2. Additionally, we select 1,034 images for 3045 instance-level and part-level segmentation to form the camouflaged image grounding segmentation task. Collectively, MMCSBench bridges the evaluation gap in existing COD datasets and establishes a unified benchmark for LVLMs to advance understanding and generalization in camouflaged environments.

Beyond evaluating LVLM capabilities, we also aim to alleviate the tedious work of camouflage image collection and improve this field. To this end, we introduce a novel and challenging task within the MMCSBench: Camouflage Efficacy Assessment (CEA). The CEA task aims to teach LVLMs to quantify the concealment of objects in any natural images, and leverage their generalization capabilities to enable the evaluation of unseen species. Furthermore, CEA can generate detailed textual descriptions regarding camouflage mechanisms (such as object-background relationships, key camouflage features), promoting the generation of multimodal camouflage resources. These capabilities also offer LVLMs a pathway to automatically evaluate the quality of existing and future camouflage datasets. Enabled by the CEA task, we further provided over 50K high-quality camouflage images, offering resources for the development of the camouflage community.

With MMCSBench, we present CamoVLM for camouflage scenes understanding. CamoVLM bridges understanding and grounding segmentation, enabling fine-grained textual characterization of camouflage environments. Details of CamoVLM in *appendix* Section D. Our key contributions are:

- We provide MMCSBench, the first comprehensive benchmark tailored for evaluating and advancing LVLM capabilities in camouflage scenes, filling a critical gap in existing resources.

- We propose the camouflage efficacy assessment task, offering a method for quantitative camouflage evaluation and paving the way for automated, large-scale collection and annotation of multimodal camouflage data. With CEA, we provide 50K+ collected high-quality camouflage images for future research.

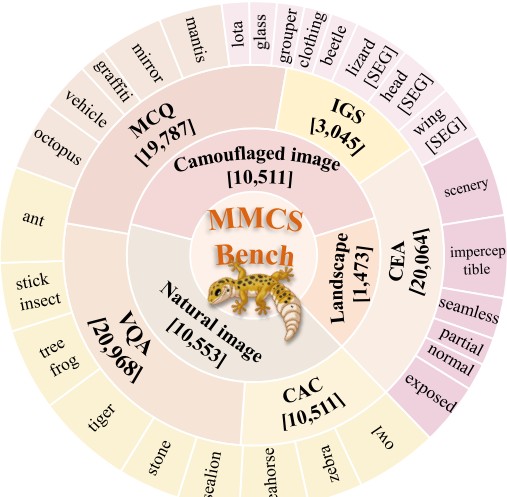

| Statistics | Number |
|---|---|
| **Basic Information** | |
| Image-Text Pairs | 76,843 |
| Training Samples | 1% |
| Testing Samples | 99% |
| **Category Information** | |
| Category Num. | 892 |
| **Text Statistics** | |
| Vocabulary Size | 14,564 |
| Avg. CAC Word Num. | 181 |
| Avg. VQA Word Num. | 38 |
| Avg. CEA Word Num. | 165 |
| Avg. IGS Word Num. | 66 |

Figure 2: **Data statistics for the MMCSBench**. The figure on the left illustrates the benchmark's data composition, showing the included tasks and the amount of data allocated to each one. The right table presents related statistics, specifically detailing information about quantity, number of categories, and text length within each task.

- Through extensive experiments on MMCSBench, we provide a comprehensive evaluation of existing LVLMs, revealing significant weaknesses in their perception of camouflage scenes and offering crucial insights to guide future research towards more capable LVLMs.

## 2 Related Work

**LVLM Benchmarks**. The rapid advancement of LVLMs underscores the critical need for effective benchmarking [7, 28, 66, 35] to identify model limitations and steer future development. While existing benchmarks like MME [13], MMBench [37], and MVBench [29] primarily evaluate perceptual and reasoning capabilities within natural scenes, they largely overlook camouflage scenarios. Such environments, often characterized by object occlusion and contextual ambiguity, present distinct challenges for LVLMs concerning both scene understanding and object detection. Furthermore, current benchmarks frequently concentrate on specific capabilities—for instance, LISA's [24] emphasis on reasoning segmentation or Video-MME's [14] focus on video understanding—rather than assessing broad, domain-general requirements. In contrast, domains such as medicine [23, 65], which involve tasks with structural parallels to camouflaged object segmentation, have seen advancements through more systematic benchmarking efforts like Fmbench [32] and MedEval [19]. Consequently, introducing the first LVLM benchmark specifically tailored to the camouflage domain is essential for addressing a significant gap and propelling progress in this field.

Table 1: **Comparison between MMCSBench and existing COD benchmarks**. We conduct a comprehensive comparison based on several aspects: the number of tasks, whether training data is included, the number of images, texts, masks, and the ability to understand camouflage.

| Benchmark | Modality | Task | Training? | Image Num. | Text Num. | Mask Num. | Answerable? |
|---|---|---|---|---|---|---|---|
| COD10K [12] | Image | 1 | ✔ | 10,000 | 0 | 5,786 | ✘ |
| NC4K [43] | Image | 1 | ✘ | 4,121 | 0 | 4,121 | ✘ |
| CHAMELEON [52] | Image | 1 | ✘ | 76 | 0 | 76 | ✘ |
| CAMO [26] | Image | 1 | ✔ | 1,250 | 0 | 1,250 | ✘ |
| COD-TAX [67] | Image, Text | 1 | ✔ | 4,040 | 4,040 | 0 | ✘ |
| OVCamo [47] | Image, Text | 1 | ✔ | 11,483 | 11,483 | 11,483 | ✘ |
| **MMCSBench** | Image, Text | 5 | ✔ | 22,537 | 76,843 | 3,045 | ✔ |

**COD and Benchmarks**. Camouflaged Object Detection (COD) [6, 8, 53, 42] aims to identify and segment hidden camouflaged objects in a given scene. Fan et al. [12] were the first to organize

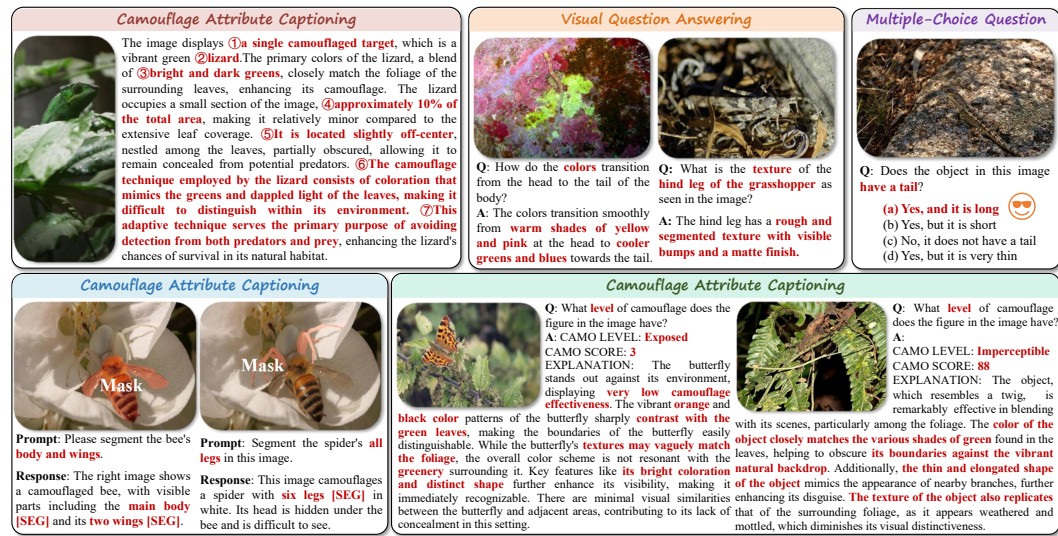

Figure 3: **Annotation presentation of MMCSBench**. For the CAC task, we manually annotate from seven different perspectives. For VQA and MCQ images, we use GPT-4V to generate fine-grained question-answer pairs based on the camouflage attribute captions. For the IGS, we not only annotate the hard-to-observe parts of objects but also generate fine-grained descriptions, where [SEG] serves as a segmentation identifier to extract segmentation tokens from LVLMs. For the CEA task, we use five levels and a 100-point scale to evaluate the level of camouflage for each image.

and propose COD10K as a benchmark for COD in training segmentation models. Since then, a number of COD benchmarks have been introduced, with the most common ones being NC4K [43], CHAMELEON [52], CAMO [26] and OVCamo [47]. As shown in Table 1, we compare MMCSBench with these datasets in terms of task, modality, and quantity. Moreover, during the construction of MMCSBench, we found that collecting camouflaged images is a highly challenging task, requiring manual inspection of a large number of images to determine whether a camouflaged object is present. This difficulty is the main reason why existing COD datasets are small and why the field has developed slowly.

## 3  MMCSBench for Camouflage Scene Understanding

### 3.1  Task Settings

In this work, we construct MMCSBench to train and evaluate existing LVLMs on their understanding of camouflage scenarios. To achieve a comprehensive and fine-grained evaluation, we design five different multimodal tasks as shown in Fig. 3: Camouflage Attribute Captioning (**CAC**), fine-grained Visual Question Answering (**VQA**), Multiple-Choice Questions (**MCQ**), Image Grounding Segmentation (**IGS**), and Camouflage Efficacy Assessment (**CEA**) for evaluating the degree of camouflage effectiveness. The detailed explanations for each task are as follows:

- **CAC**: Given a camouflage image, LVLMs are required to identify and describe the basic attributes of the camouflaged object, such as quantity, category, size, color, and location. Additionally, they need to understand the relationship between the object and its environment, as well as the camouflage strategies adopted and the purpose of the camouflage.

- **VQA**: Unlike conventional VQA tasks [35, 37], VQA in camouflage scenes requires LVLMs to provide accurate answers to fine-grained questions. For example, a question should be "How many claws of the camouflaged object are gripping the branch in the image?" rather than "What is the category of the camouflaged object?", as the latter has already been addressed in CAC task.

- **MCQ**: Similar to VQA, MCQ in camouflage scenes also requires fine-grained cognition. However, the key difference is that MCQ provides four answer choices. Additionally, LVLMs are required to provide a detailed explanation for their chosen option.

- **IGS**: Compared to text generation tasks, pixel-level segmentation tasks require a more detailed understanding of camouflage scenes. In the proposed IGS task, we provide part-level masks along with corresponding prompts, such as `"Segment the most camouflaged part of the spider"`, with the expected response being `"Its six legs [SEG]"`.

- **CEA**: Given any image, whether camouflaged or not, the CEA task requires LVLMs to assess the camouflage level of the object using five categories: `Exposed`, `Normal`, `Partial`, `Seamless`, and `Imperceptible`, as shown in Fig. 3. Additionally, they must quantify the camouflage score on a scale of 0-100 (the higher the more camouflaged) and provide a detailed explanation about the camouflaged object and its surrounding environment.

## 3.2 Benchmark Construction

**Images Collection**. Searching for and collecting camouflaged images from numerous natural images is extremely tedious and time-consuming. We would like to express our gratitude to the work of previous researchers [12, 43, 52, 26], whose contributions provided a wealth of high-quality camouflaged object images that supported the construction of MMCSBench and also motivated us to develop an automated collecting method. Specifically, we collected and selected a total of 10,511 camouflaged images from the COD10K [12], NC4K [43], CHAMELEON [52], and CAMO [26] datasets, covering various scenes and different categories of objects. Furthermore, for the CEA task, we incorporate 10,553 natural images from the DUTS dataset [58] and 1473 landscape images to mitigate data bias caused by the high camouflage levels in the collected COD datasets.

**CAC, VQA and MCQ Tasks Generation**. Image understanding within LVLMs centers on image captioning, a fundamental task for VLM [57, 3]. For this, we first construct detailed captions for camouflaged images. Given the poor performance of the existing LVLMs on camouflage scenes, we manually annotate these captions, ensuring fine-grained textual representations that emphasize object details and their environmental context. The accuracy of these captions is vital, as they form the foundational data for other tasks. Each caption undergoes collaborative refinement by three expert annotators, resulting in 10,511 high-quality camouflaged attribute descriptions for the CAC task. Building upon these captions, we establish datasets for VQA and MCQ tasks. Unlike conventional datasets, our approach necessitates fine-grained questions designed to deepen LVLMs' understanding of camouflage scenes. Using images and their corresponding captions as input, we employ GPT-4V [1] to generate diverse VQA and MCQ questions. For each image, we generate 3 VQA pairs and 3 MCQ sets, which then undergo rigorous filtering to remove inaccurate, ambiguous, and logically flawed questions. These filtered questions are then answered by 5 camouflage experts. Integrating their answers yields 20,968 VQA pairs and 19,787 MCQ sets. Fig. 3 illustrates examples of these annotations. We provide more MMCSBench examples in Section B of the *appendix*.

**IGS Task Generation**. Unlike CAC, VQA, and MCQ tasks, the IGS task requires specifying segmentation prompts, text responses, and pixel-level masks. Therefore, we select 900 images containing multiple instances from 10,511 images for separate instance-level annotation, resulting in 2,012 instance masks. Additionally, we select 534 images containing fine-grained objects for part-level annotation (e.g., "head, claws, tail"), obtaining 1,033 part masks, and generate corresponding descriptions using GPT-4V. Similar to LISA [24] and GeoPixel [2], we insert segmentation identifiers `[SEG]` into the responses, enabling the extraction of segmentation tokens from LVLM's output via identifier ID. These tokens are then fed into the model's built-in segmentation network.

**CEA Task Generation**. The proposed CEA task aims to teach LVLMs to recognize the degree of camouflage of objects in any given image, addressing the issue of insufficient training data in the camouflage domain. A prerequisite for this task is to establish a quantitative criterion for each image. Based on the work of Lamdouar et al. [25], we propose to use multi-dimensional feature similarity and edge visibility to assess the camouflage score. This score $C_s$ is calculated from the color ($S_c$), texture ($S_t$), and edge fusion ($S_e$) similarities between foreground and background. These metrics, based on [25], capture key camouflage aspects of the given image. Specifically, the $S_c$ is:

$$S_c(I, M) = 1 - \sum_{h,s,v} \sqrt{H_T(h, s, v) \cdot H_B(h, s, v)} \cdot \left(1 - \frac{\|H_T - H_B\|_2}{\|H_T\|_2 + \|H_B\|_2}\right)^{-1}, \tag{1}$$

where $H_T(h, s, v)$ and $H_B(h, s, v)$ denote the normalized HSV color histograms of the foreground and background, respectively, and $\| \cdot \|_2$ represents the $L_2$ norm used to measure the Euclidean

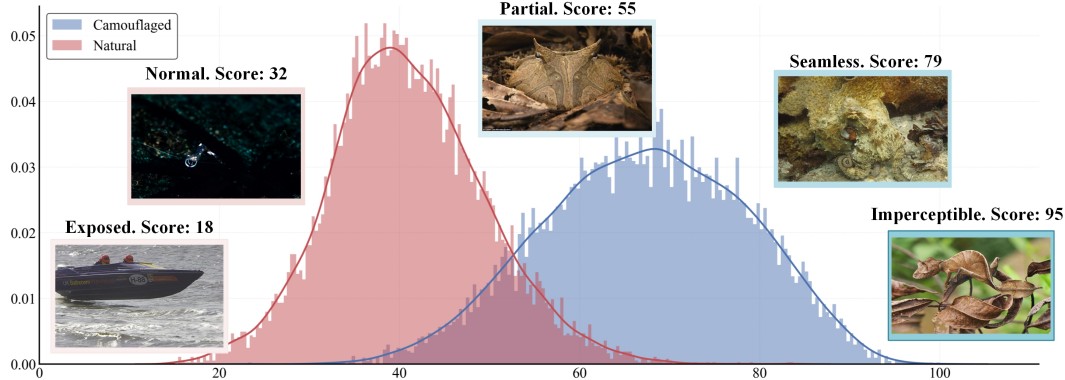

Figure 4: **Examples of camouflage scores for different images and the overall distribution of camouflage scores for images in MMCSBench**. The x-axis represents the computed camouflage score $C_s$. We visualize the score distributions separately for camouflaged images and natural images.

distance between histograms. For texture similarity $S_t$, we employ Local Binary Patterns (LBP) to capture micro-texture differences:

$$S_t(I, M) = 1 - \frac{1}{2} \sum_{i=1}^{n_{bins}=30} (P_T(i) - P_B(i))^2 \cdot \log\left(\frac{P_T(i) + P_B(i)}{2}\right), \qquad (2)$$

where $P_T$ and $P_B$ are the normalized LBP histogram distributions. The formula computes a divergence score comparing these two distributions to quantify texture dissimilarity. To evaluate the degree of edge integration between the object and the background, we introduce edge fusion similarity $S_e$:

$$S_e(I, M, E) = \sum_{c \in \{L,a,b\}} \left( 1 - \tanh\left( \frac{1}{|\mathcal{R}_f|} \sum_{(x,y) \in \mathcal{R}_f} (I_{\text{Lab}}^c(x,y) - \mu_{\mathcal{R}_f}^c)^2 \right) \right), \qquad (3)$$

where $\mathcal{R}_f$ denotes the fusion region formed by dilating the object edges into the background in the CIELab color space. The score $C_s$ is a weighted sum of the three scores: $C_s = \alpha S_c + \beta S_t + \gamma S_e$ (where weights $\alpha$, $\beta$, and $\gamma$ are 0.5, 0.3, and 0.2; weight analysis in *appendix* Section C.4).

Since regression learning is more challenging for LVLMs, we classify the computed camouflage scores (0-100) into five distinct levels, as shown in Fig. 4. This figure presents the score distributions for camouflage and natural images, where the x-axis indicates the camouflage score and the y-axis represents the proportion of samples. The score distributions not only reveal clear discriminability between camouflaged and natural images but also exhibit regular trends (e.g., approximating normality), suggesting consistency in our scoring mechanism. Moreover, in the CEA task, we require LVLMs to simultaneously learn the target's camouflage level and specific score. They must also provide detailed descriptions relating the object's camouflage to its boundary region and the overall environment, thus fostering the application of CAC within the field of multimodal camouflage.

Notably, training the CEA task solely with images containing confirmed objects is inadequate. This causes the LVLM to strongly hallucinate, mistakenly perceiving highly camouflaged objects within pure natural images and assigning them high camouflage scores. To mitigate this, we gathered 1,473 landscape images as negative examples for training together. This helps the LVLM first determine the presence or absence of highly camouflaged objects before assessing the camouflage level. Corresponding captions for these images are `"This is a landscape image, with no particularly distinctive objects"`, followed by scenery details. Examples in Section C.2.

**CEA with 50K+ Camouflage Images**. Using Qwen2.5-VL-72B [4] fine-tuned on the CEA task, we performed automated camouflage image detection and collection on the COCO [31], WildFish [73], iNat [56], and AK [44] datasets, obtaining 54,468 camouflaged images and their text captions. Overall, we are the first work to achieve automated collection of camouflage images, providing valuable experience and data for the camouflage community. The samples are in Section C.3.

Table 2: **Performance of different LVLMs on MMCSBench**. 'Human Eval.' indicates results from professional annotators. ↑ denotes higher is better, ↓ denotes lower is better. The best values are **bolded** and the second-best are underlined.

| LVLM | CAC | | | VQA | | | MCQ | | | CEA | | |
|---|---|---|---|---|---|---|---|---|---|---|---|---|
| | BLEU↑ | METEOR↑ | RougeL↑ | BLEU↑ | METEOR↑ | RougeL↑ | ACC↑ | BLEU↑ | METEOR↑ | ACC↑ | MAE↓ | BLEU↑ |
| *Models without Supervised Fine-tuning* | | | | | | | | | | | | |
| ●GPT-4.1 [1] | 12.39 | 25.82 | 22.65 | 13.29 | 23.81 | 22.05 | 72.75 | 16.89 | 38.66 | 62.51 | 10.93 | 12.07 |
| ●Gemini2.5-Pro (05-06) [55] | **14.82** | **33.28** | **30.88** | 15.83 | 31.93 | 34.08 | 75.38 | 19.73 | 39.89 | 66.88 | 8.67 | 15.90 |
| ●Qwen2.5-VL-72B [4] | 10.87 | 23.36 | 22.08 | 11.70 | 23.07 | 21.15 | 73.08 | 14.08 | 40.96 | 60.87 | 12.66 | 11.55 |
| *Models with Supervised Fine-tuning (Grouped by Size)* | | | | | | | | | | | | |
| ●OVIS2-2B [41] | 11.54 | 33.11 | 26.85 | 14.62 | 38.43 | 37.36 | 58.82 | 12.92 | 26.46 | 64.72 | 13.46 | 15.47 |
| ●InternVL3-2B [72] | **17.06** | 40.04 | **31.25** | 19.00 | 44.25 | **41.71** | 73.88 | 25.64 | 43.80 | 68.05 | 10.24 | 17.88 |
| ●Qwen2.5-VL-3B [4] | 15.69 | **40.57** | 30.35 | **19.06** | **44.69** | 41.45 | 72.41 | 24.99 | 43.08 | 69.34 | 8.91 | 19.27 |
| ●Molmo-7B [11] | 19.13 | 42.85 | 35.27 | 20.61 | 46.22 | 43.89 | 76.42 | 26.38 | 44.57 | 68.53 | 7.80 | 18.55 |
| ●LLAVA1.5-7B [33] | **19.87** | **43.46** | **36.34** | 21.32 | **46.78** | 44.40 | 77.28 | 26.50 | **45.24** | 65.31 | 9.86 | 17.20 |
| ●LLAVA-NEXT-8B [34] | 19.84 | 43.44 | 36.29 | **21.33** | **46.78** | 44.41 | **77.56** | 26.99 | 45.23 | 66.26 | 8.72 | 17.42 |
| ●InternVL2.5-8B [9] | 18.06 | 42.14 | 34.30 | 20.74 | 46.43 | 44.21 | 76.58 | 26.59 | 44.80 | 71.88 | 6.59 | 19.34 |
| ●InternVL3-8B [72] | 18.44 | 42.15 | 33.30 | 20.68 | 46.45 | 44.05 | 76.02 | 26.21 | 44.63 | 72.36 | 6.53 | 19.69 |
| ●Qwen2.5-Omni-7B [63] | 18.23 | 42.77 | 34.09 | 19.56 | 45.22 | 42.13 | 75.22 | 26.31 | 44.02 | 72.36 | 6.48 | 20.18 |
| ●Qwen2.5-VL-7B [4] | 18.09 | 42.52 | 33.87 | 19.25 | 45.10 | 41.83 | 75.20 | 25.71 | 43.61 | **72.84** | **6.32** | 20.25 |
| ●GLM-4V-9B [17] | 18.53 | 42.75 | 35.27 | 20.61 | 46.32 | 43.59 | 76.42 | 26.38 | 44.57 | 70.25 | 7.53 | 18.55 |
| ●LLAVA1.6-34B [33] | 20.31 | **45.68** | **38.09** | 22.36 | 46.98 | 45.23 | **81.23** | 27.04 | **46.31** | 73.26 | 5.69 | 20.35 |
| ●InternVL3-38B [72] | 19.42 | 44.10 | 35.82 | 25.17 | 49.32 | 48.18 | 80.72 | 26.92 | 45.28 | 72.17 | 6.43 | 19.49 |
| ●Qwen2.5-VL-32B [4] | 19.64 | 44.29 | 36.07 | 20.03 | 45.68 | 42.32 | 78.92 | 26.02 | 43.88 | **74.22** | 5.74 | 20.52 |
| ●NVLM-D-72B [10] | **21.65** | **47.12** | **38.45** | 21.03 | 47.25 | 46.38 | 81.15 | 26.45 | 50.27 | 74.62 | 5.47 | 20.88 |
| ●LLAVA-OV-72B [27] | 20.85 | 46.32 | 38.18 | 20.76 | 46.25 | 45.92 | 83.45 | 26.67 | 51.38 | 76.28 | 5.25 | **22.07** |
| ●InternVL3-78B [72] | 21.07 | 46.59 | 37.94 | **26.32** | **50.27** | **49.83** | **83.37** | **27.08** | **56.84** | **76.94** | **4.98** | 21.49 |
| ●Qwen2.5-VL-72B [4] | 21.51 | 46.90 | 38.27 | 20.29 | 46.82 | 44.01 | 82.78 | 26.74 | 54.72 | 76.31 | 5.02 | 21.52 |
| Human Eval. | - | - | - | - | - | - | 85.32 | - | - | 79.56 | 3.62 | - |

# 4 Experiments

## 4.1 Experimental Settings

**Benchmark Settings**. We aim to evaluate the understanding of LVLMs in camouflage scenes. However, directly evaluating these models on MMCSBench yields unsatisfactory results, as shown in Table 2. This suboptimal performance stems partly from the poor ability of these LVLMs to recognize camouflage, and partly from difficulties in adhering to the required output formats for text. Therefore, we fine-tune these LVLMs using 1% of the MMCSBench data (sampled with balance across five CEA levels) to tailor them for camouflage images, and subsequently test their performance on the remaining 99% data. This is also advantageous for evaluating the effectiveness of LVLMs on unseen camouflage categories and environments. Details about the training and testing, as well as higher training percentages (10%, 50%) performance, can be found in Section E.1 and E.4 of the *appendix*. The 50K+ camouflage images collected via the CEA task can be seen in Section C.3.

**Metrics**. For text generation tasks such as CAC, VQA, MCQ, and CEA, we adopt standard evaluation metrics including BLEU-4 [48], METEOR [5], and RougeL [30] to assess the quality of the generated text. Specifically, for MCQ and CEA tasks, we additionally use accuracy to evaluate the correctness of option selection and camouflage level. For the predicted camouflage score, we employ the Mean Absolute Error (MAE) to measure the deviation from ground truth values. For IGS task, we use BLEU-4, mean Intersection over Union (mIoU) [39] and Recall as evaluation metrics.

**Baseline LVLMs**. From the perspective of task categorization, MMCSBench includes text generation tasks like CAC, VQA, MCQ, and CEA, as well as the segmentation task IGS. Since most LVLMs do not have grounding segmentation capabilities, we divide the evaluation into two parts. For text generation tasks, the evaluated LVLMs include GPT-4.1 [1], Gemini2.5-Pro [55], Qwen2.5-VL [4], OVIS2 [41], LLAVA1.5(1.6) [33], LLAVA-NEXT [34], InternVL3 [72], Molmo[11], Qwen-Omni [63], GLM-4V [17], NVLM-D [10] and LLAVA-OV [27]. For the IGS task, we select models specifically designed for grounding segmentation, including LISA [24], PixelLM [50], GeoPixel [2], GlaMM [49], and SegLLM [61]. These models are fine-tuned on MMCSBench training set and compared against the proposed CamoVLM. Details can be found in Section E of *appendix*.

**Human Evaluation**. In this work, we conduct a human evaluation to compare the actual performance of existing LVLMs in camouflage scenarios. Specifically, we ask five expert evaluators to select answers for the options in both MCQ and CEA tasks, and use the average performance of the five evaluators as the human evaluation performance. The results "Human Eval." are shown in Table 2.

**Performance on Text Generation Tasks**. Table 2 compares how different LVLMs performed on the four text generation tasks. It's clear that supervised fine-tuning gives a major boost over zero-shot models like GPT-4.1. Among the zero-shot group, Gemini2.5-Pro generally came out on top, but still fell well short of the fine-tuned models. Even the small 2-3B models improved on zero-shot results, with InternVL3-2B and Qwen2.5-VL-3B being the front-runners here, trading the lead depending on the metric. In the 7-8B tier, the LLAVA models (1.5 and NEXT) handled generation tasks and MCQ accuracy well, while Qwen2.5-VL-7B stood out on the CEA metrics (ACC, MAE, BLEU). The InternVL models (2.5 and 3) were also competitive, notably grabbing the second-best CEA MAE. Moving up to 32-38B models, LLAVA1.6-34B showed strength in CAC and MCQ, InternVL3-38B led in VQA, and Qwen2.5-VL-32B did well on CEA accuracy and BLEU. The largest 72B+ models delivered the top scores overall. NVLM-D-72B was strongest in CAC; InternVL3-78B excelled across VQA and MCQ text generation, also hitting the best CEA accuracy (76.94%) and MAE (4.98); and LLAVA-OV-72B took the top spot for MCQ accuracy (83.45%) and CEA BLEU. Qwen2.5-VL-72B often landed in second place. However, even these top-performing models lag significantly behind human evaluations (85.32% MCQ Acc, 79.56% CEA Acc), indicating substantial room for improvement for current LVLMs in understanding complex camouflage scenarios.

**Performance on IGS Task**. Existing grounding methods struggle in camouflaged scenes due to high-resolution input requirements and no specific multimodal alignment design. Thus, we introduce CamoVLM. Its dedicated segmentation capability enhances feature alignment via a lookup-matching strategy. This strategy generates tailored attention masks, enabling precise instruction following and accurate grounding in challenging camouflaged environments.

Table 3: **Performance of different grounding segmentation models on the MMCSBench IGS task**. The results shown are for methods that have all been fine-tuned.

| Model | MLLM | BLEU↑ | mIoU↑ | Recall↑ |
|---|---|---|---|---|
| LISA [24] | LLAVA1-7B | 17.80 | 60.08 | 48.09 |
| PixelLM [50] | LLAVA1-7B | 17.92 | 60.27 | 48.74 |
| GeoPixel [2] | Xcomposer2-7B | 18.98 | 64.26 | 50.52 |
| GLaMM [49] | LLAVA1.5-7B | 18.53 | 62.05 | 49.73 |
| SegLLM [61] | LLAVA1.5-7B | 18.61 | 62.80 | 50.19 |
| **CamoVLM** | Qwen2.5-VL-7B | **19.36** | **65.81** | **52.88** |

Table 3 presents the performance of various grounding segmentation models on the challenging MMCSBench IGS task. All models were fine-tuned on the training subset. The results show that our proposed CamoVLM consistently outperforms the compared existing methods across all evaluation metrics: BLEU (for text generation), mIoU, and Recall (for segmentation accuracy). Specifically, CamoVLM achieves 19.36 BLEU, 65.81 mIoU, and 52.88 Recall. Compared to the next best performing model, GeoPixel, CamoVLM demonstrates a notable improvement, leading by 0.38 in BLEU, 1.55 in mIoU, and 2.36 in Recall. This highlights CamoVLM's improved capability in both interpreting fine-grained instructions and accurately segmenting camouflaged objects or parts within complex scenes.

Table 4: **Generalization performance of CamoVLM on IGS task**. Performance (BLEU, mIoU, Recall) is shown for test sets with N unseen object categories (N=10, 20, 30, 40, 50), compared to a baseline on seen categories.

| Setting | BLEU↑ | mIoU↑ | Recall↑ |
|---|---|---|---|
| BL | 19.78 | 66.26 | 53.07 |
| N=10 | 19.37 | 66.06 | 52.94 |
| N=20 | **20.06** | **66.39** | **53.21** |
| N=30 | 19.30 | 66.00 | 52.79 |
| N=40 | 19.51 | 66.09 | 52.87 |
| N=50 | 19.26 | 65.83 | 52.80 |

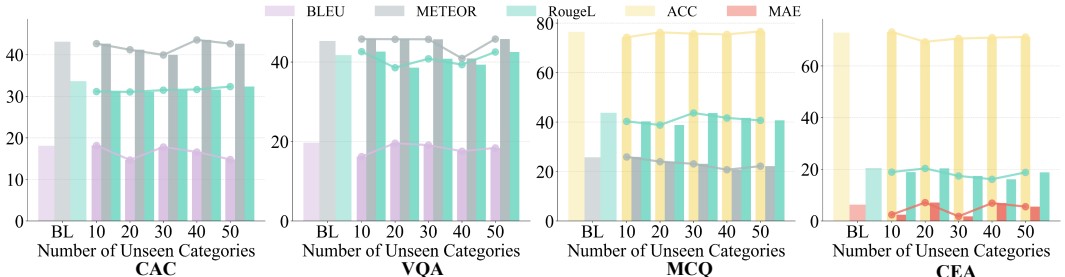

Figure 5: **Generalization performance of Qwen2.5-VL-7B on CAC, VQA, MCQ, and CEA tasks**. "BL" is the baseline performance on seen categories.

## 4.2 More Experiments

**Generalization to Unseen Camouflaged Categories**. Evaluating the generalization capability of LVLMs after fine-tuning on MMCSBench in camouflage scenarios is crucial, especially their ability to recognize object categories not previously encountered. To this end, we design an experiment to explore the understanding and grounding performance of Qwen2.5-VL-7B and our CamoVLM when encountering unseen camouflaged object categories. Specifically, for the BaseLine (BL), we select 50 classes from MMCSBench for training and validation, meaning the baseline's validation set consists entirely of seen categories. Subsequently, we evaluate their performance on different validation sets composed of 10, 20, 30, 40, or 50 unseen categories. Fig. 5 and Table 4 respectively show the models' understanding and grounding capabilities on these different sets of unseen categories.

From Fig. 5 and Table 4, we can observe that as the number of unseen categories increases, there is no obvious trend of performance degradation. This indicates that integrating LVLMs for understanding camouflage scenes is effective and can improve their performance in unknown scenarios. Notably, there is a significant variation in the number of samples across different CEA levels. To address this, we employed a camouflage level balancing strategy during training, ensuring an equal number of training samples for each camouflage level.

**Validation of CEA for Automated Camouflage Image Discovery**. The primary objective of CEA task is to identify camouflaged images within extensive datasets, while assessing its robustness across unseen object categories. Thus, we evaluate LVLMs fine-tuned on MMCSBench across the COD datasets PCOD [60] and PlantCamo [64], alongside the natural image datasets COCO [31] and ImageNet-S [15]. These datasets, encompassing a diverse range of object categories, serve as a robust benchmark to effectively assess the capabilities of LVLMs for the CEA task.

Table 5: **Evaluating fine-tuned InternVL3 models on the CEA task using external datasets**. Metrics include MAE for camouflage score prediction and error rates for identifying Seamless (Err @4) and Imperceptible (Err @5) levels.

| LVLM | PCOD [60] | | | CamoPlant [64] | | | COCO [31] | | | ImageNet-S [15] | | |
|---|---|---|---|---|---|---|---|---|---|---|---|---|
| | MAE↓ | Err @4↓ | Err @5↓ | MAE↓ | Err @4↓ | Err @5↓ | MAE↓ | Err @4↓ | Err @5↓ | MAE↓ | Err @4↓ | Err @5↓ |
| InternVL3-1B | 11.28 | 9.28% | 3.28% | 11.02 | 10.05% | 3.58% | 11.52 | 9.31% | 4.15% | 12.53 | 9.50% | 3.62% |
| InternVL3-2B | 10.31 | 8.72% | 2.33% | 11.46 | 7.98% | 3.42% | 11.03 | 8.63% | 2.36% | 10.92 | 8.32% | 2.48% |
| InternVL3-8B | 6.11 | 6.62% | 1.74% | 6.36 | 7.69% | 2.41% | 5.95 | 6.81% | 2.04% | 6.30 | 7.77% | 1.53% |
| InternVL3-38B | 5.79 | 6.02% | 0.95% | 6.24 | 6.51% | 1.19% | 5.51 | 5.32% | 0.89% | 5.89 | 6.48% | 1.26% |
| InternVL3-78B | 5.21 | 4.20% | 0.27% | 5.60 | 4.32% | 0.48% | 4.98 | 4.14% | 0.35% | 5.02 | 3.90% | 0.37% |

In Table 5, we present the performance of the InternVL3 series of models on these four datasets. Notably, we specifically highlight the error rates of these models on `Level 4 Seamless` and `Level 5 Imperceptible` (Err @4, and Err @ 5), to align with our primary goal of detecting camouflaged images. As we can see, the performance consistently improves with model scale, and critically, larger models like InternVL3-78B achieve high accuracy in identifying the most challenging Seamless (Level 4) and Imperceptible (Level 5) camouflage instances. Notably, error rates for detecting Level 5 camouflage (Err @5) were below 1% on dedicated camouflage datasets and remained low ( 1%) even on large-scale natural datasets like ImageNet-S. These results strongly validate CEA's potential

as an efficient tool for automatically identifying and assessing camouflaged images within extensive datasets, thereby facilitating the expansion of data resources for the camouflage community.

## 5   Conclusion

**Conclusion**. We introduced MMCSBench, the first multimodal benchmark for LVLM camouflage understanding. With over 75,000 novel image-text pairs across five tasks, it addresses a key evaluation gap. We also proposed the CamoVLM baseline. Experiments show fine-tuning improves LVLM performance, yet a significant gap to human accuracy persists, underscoring the challenge. MMCSBench offers a valuable resource to drive research in fine-grained visual understanding and advance LVLM capabilities.

## Acknoledgements

This work was funded by the National Natural Science Foundation of China under grants 62475016, the Beijing Natural Science Foundation under Grant L252142, and the Science and Technology on Electromechanical Dynamic Control Laboratory Funding under grant 6142601012402.

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

# Appendix for MMCSBench

- **Section A (Motivation)**. *More Results of Representative LVLMs on Camouflage Scenes*: We demonstrate the performance of representative existing Large Vision-Language Models (LVLMs) in camouflage scenarios to highlight the necessity of constructing a dedicated camouflage dataset.
- **Section B (Contribution 1 Benchmark)**. *More MMCSBench Samples*: We present more image samples from the MMCSBench dataset, accompanied by their corresponding textual descriptions and segmentation mask labels.
- **Section C (Contribution 2 CEA Task)**. *CEA Task Supplement*: We showcase more images at different levels to demonstrate the grading effect of CEA. Furthermore, we explain the hyperparameters in $S_c$ and introduce the 50K+ high-quality camouflage images obtained via CEA, further proving the practicality and applicability of the proposed CEA task.
- **Section D (CamoVLM Baseline)**. *CamoVLM for Camouflage Scene Understanding*: We offer a detailed description of the proposed CamoVLM model and present its segmentation performance. However, we reiterate that CamoVLM is not positioned as the core technical innovation of this paper.
- **Section E (Experiment)**. *Implementation Details of Training and Testing*: We provide further details on training and testing, including LoRA settings, prompts, and hyperparameter settings.

## Broader Impacts

MMCSBench provides a valuable benchmark and dataset to accelerate AI research in fine-grained visual understanding, particularly for challenging camouflage scenes. This work fosters innovation towards more capable AI systems, with no foreseen negative societal consequences.

## Limitations

We acknowledge certain limitations. First, incorporating camouflaged images into pretraining rather than fine-tuning might yield better results; however, considering training efficiency, we focused on fine-tuning and still achieved strong performance. Second, CamoVLM details are placed in the appendix to keep the main text focused on the **MMCSBench**. Third, the quantitative CEA score uses specific features and weights; exploring alternatives is a direction for future work.

## A    More Results of Representative LVLMs on Camouflage Scenes

In Fig. 6 and Fig. 7, we show more results of existing LVLMs (Qwen-Max, GPT-4.1, Gemini2.5-Pro) on camouflage scenarios. As can be seen from Fig. 6, it remains difficult for these LVLMs to simply identify camouflaged targets without any prompts; not only might they fail to identify the camouflaged target, but they might still provide incorrect answers even after being prompted about the existence of a camouflaged target. For multiple-choice questions, because information about the target's existence and category is provided in the question, these LVLMs are able to provide the correct answer with some probability, which is consistent with our numerical results in Table 2.

Overall, existing large-scale LVLMs struggle to achieve the same performance in recognizing camouflage scenarios as they do on natural images, thus leading to weakened reasoning capabilities. The introduction of MMCSBench can improve LVLMs' understanding of camouflage scenarios and provides a new direction for the future development of LVLMs.

## B    More MMCSBench Examples

In this section, we present further examples of samples and their corresponding annotations from the MMCSBench dataset. These examples are organized according to task, as illustrated in Fig. 8, which showcases instances for both Camouflaged Attribute Captioning (CAC) and Visual Question Answering (VQA) tasks.

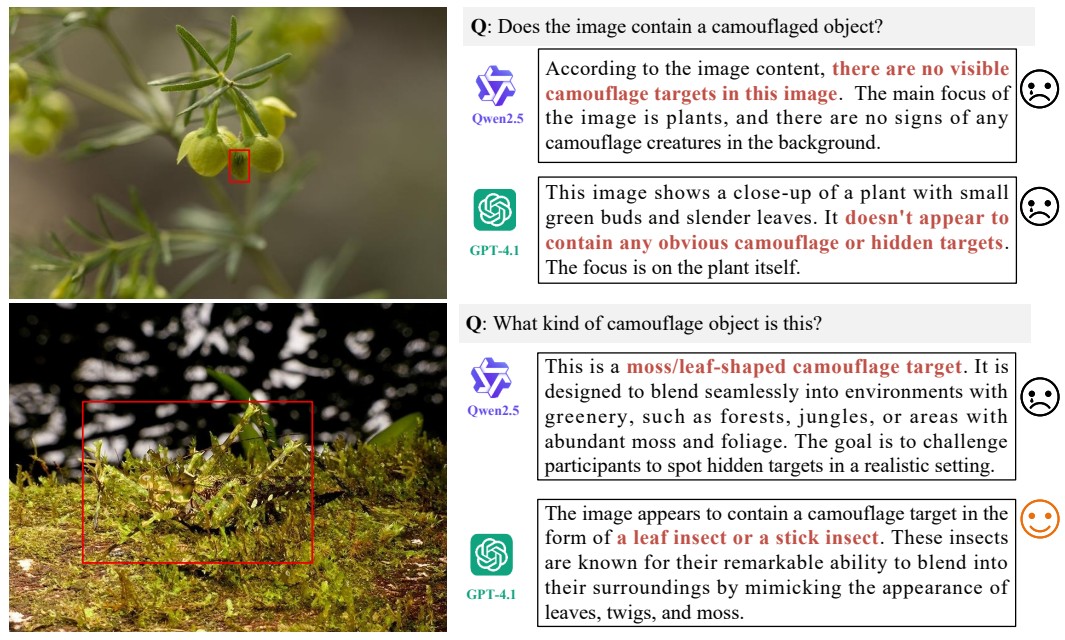

Figure 6: **Poor performance of mainstream LVLMs in camouflage scenarios**. We evaluate GPT-4.1 and Qwen2.5-Max on their ability to describe objects in camouflage scenes.

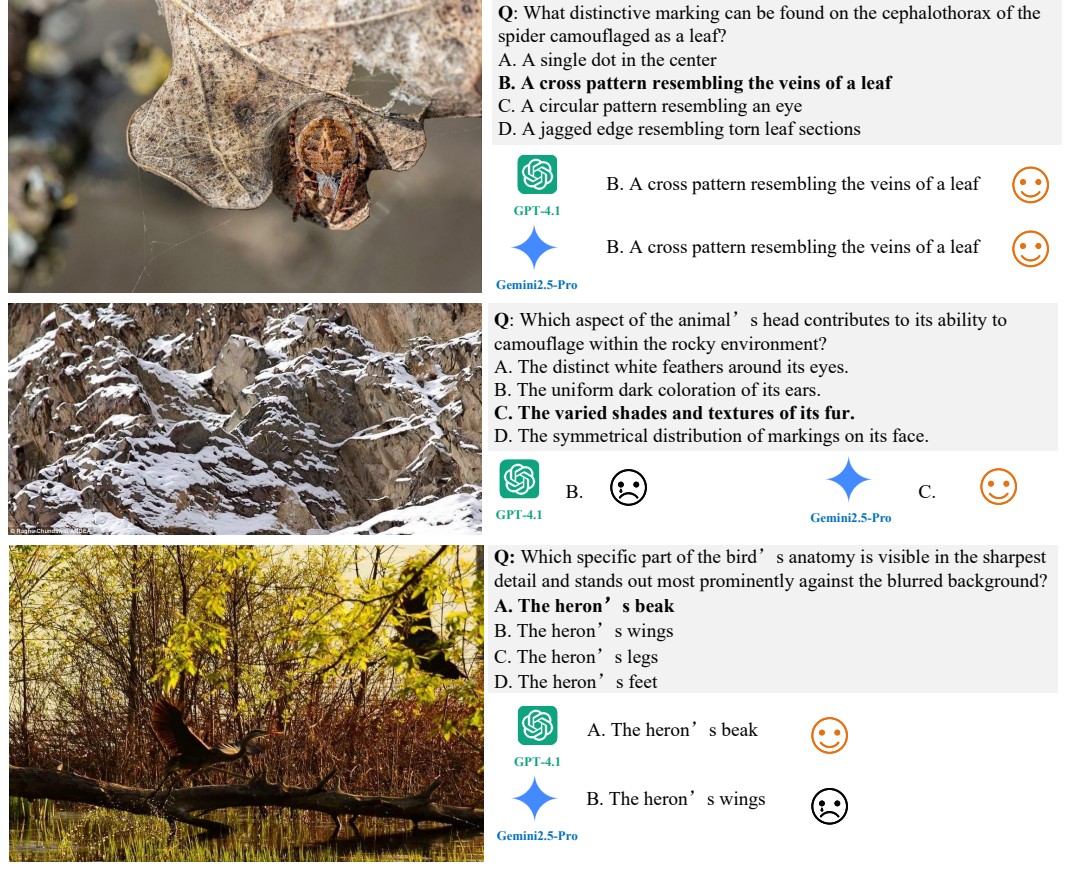

Figure 7: **Poor performance of mainstream LVLMs in camouflage scenarios**. We evaluate GPT-4.1 and Gemini2.5-Pro on their performance on multiple-choice questions in camouflage scenes. Notably, these LVLMs are able to improve their results when explicit cues are included in the prompt.

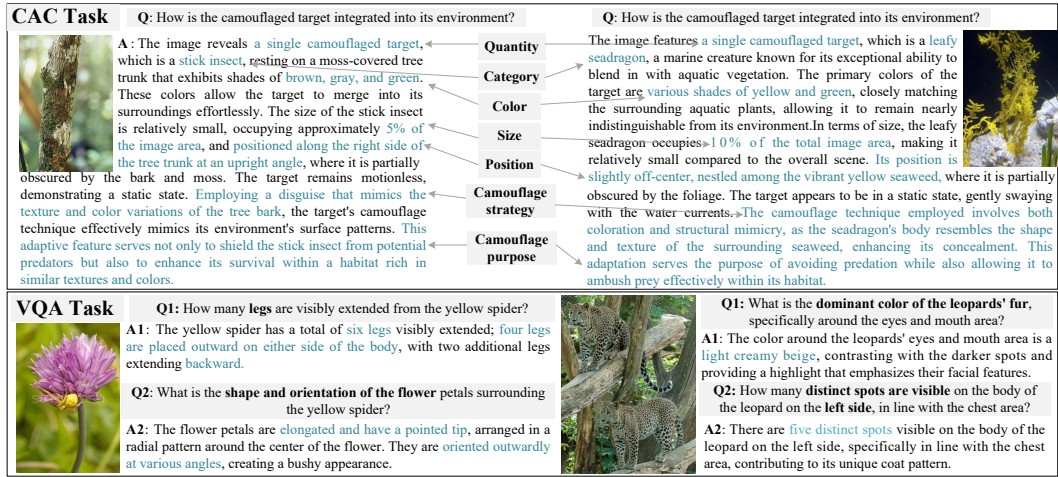

Figure 8: **Examples of CAC and VQA tasks in MMCSBench**. For the CAC task, our way of asking questions is fixed.

## B.1 Examples for CAC Task

For the CAC task, annotations are meticulously produced through extensive manual effort. Recognizing the limitations of automated methods in capturing the subtleties of camouflage, we relied on human annotators to ensure high fidelity. A rigorous annotation protocol was established, mandating detailed descriptions of the camouflaged object(s) based on seven key aspects: quantity present, object category, estimated size, dominant colors, spatial position within the frame, the specific camouflage technique employed (e.g., background matching, mimicry, disruptive coloration), and the inferred camouflage purpose (e.g., predator avoidance, ambush predation). This multi-faceted approach was designed to yield a comprehensive characterization of the camouflaged entity. Furthermore, beyond describing the object itself, descriptions detailing its interaction with the immediate surrounding environment were explicitly mandated. This involved noting how the object integrates with specific background elements (like foliage, bark, or substrate), utilizes shadows, or adopts postures that enhance its concealment, thereby providing crucial context about how the camouflage functions. Our curated collection comprises a substantial total of 10,511 unique camouflaged images, resulting in 10,511 corresponding image-text pairs specifically generated for the CAC task. Each pair serves as a rich, fine-grained textual representation intended to ground the complex visual characteristics of the camouflaged subjects.

## B.2 Examples for VQA Task

For the VQA task, we utilized the comprehensive descriptions generated during the CAC annotation phase as input prompts. These prompts were subsequently processed by GPT-4V to automatically generate challenging, fine-grained question-answer pairs. An exemplary question generated through this process, shown in the figure, is: `How many legs are visibly extended from the yellow spider?`. Accurately answering such questions necessitates that LVLMs possess a sophisticated, pixel-level understanding of the visual content.

## B.3 Examples for MCQ Task

Fig. 9 showcases data samples from the Multiple-Choice Question (MCQ) and Image Grounding Segmentation (IGS) tasks integrated within the MMCSBench. In the MCQ component, we maintain a focus on the intricate details of target objects, moving beyond simplistic questions. Furthermore, these questions are specifically designed to evaluate the sophisticated reasoning capabilities of Large Vision-Language Models (LVLMs). For instance, one question asks, `which specific part of the lizard's anatomy is prominently displayed?`. Successfully answering this necessitates that the LVLM not only comprehends the visual information but also infers the most conspicuous anatomical feature of the lizard within the image context. Another example involves identifying the

**MCQ Task**

**Q**: In the image, which **specific part of the lizard's anatomy is prominently displayed** while it is positioned in the branches, and what unique feature is visible on that part?
A. The tail, showcasing a gradient coloring transition.
B. The foot, exhibiting claw details with a slight curvature.
C. The eye, displaying a distinct reflective quality.
D. The dorsal ridge, highlighting elongated scales that catch the light.

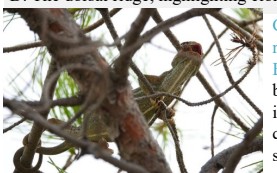

C. The eye, displaying a distinct reflective quality.
Explanation: The lizard's eye can be seen vividly focused on in the image, exhibiting a reflective quality that highlights its glossy surface. This reflective characteristic is achieved through a combination of the eye's structure and the surrounding lighting, making it stand out against the more muted colors of the lizard's body. The eye is also slightly open, emphasizing its alertness and enhancing the overall detail captured in the pixel-level examination of the image.

**Q**: Which specific feature of the octopus's body is **most prominently** camouflaged against the coral background?
A. The shape of its head
B. The color pattern on its body
C. The position of its eyes
D. The arm positioning

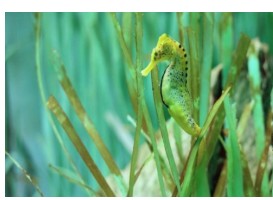

B. The color pattern on its body
Explanation: The octopus's body, particularly its grayish, mottled color pattern, closely matches the sandy, coral-laden seabed, making it difficult to distinguish from surrounding environment. Indispensable for its camouflage, this specific color pattern allows the organism to achieve a remarkable, flawless integration with its background environment. This ability to seemingly vanish is not just advantageous, it represents a crucial survival mechanism essential for its existence.

**IGS Task**

**Q:** Segment the fish's head and fin **on the right**.

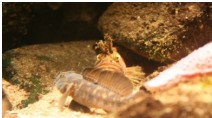
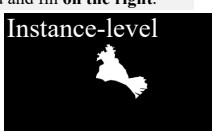
Instance-level

**A**: The image depicts the upper section of a camouflaged fish, specifically its head [SEG] and a portion of its fin [SEG]. The fin is extended laterally and partially visible, while the head showcases the pronounced features such as eyes and dorsal protrusions that were outlined in the full image on the left.

**Q:** Segment the gecko's **head**.

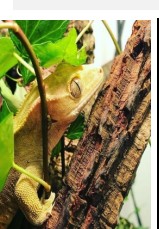
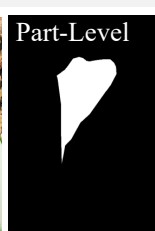
Part-Level

**A**: The image shows the head [SEG] of a camouflaged gecko visible amidst its natural bark environment. The head is the focal point, featuring a textured surface with a prominent eye and part of the snout. The rest of the body is obscured, making only the head [SEG] clearly distinguishable.

Figure 9: **Examples of MCQ and IGS tasks in MMCSBench**. For the MCQ task, we require LVLMs to provide detailed explanations for their options. For the IGS task, we specifically annotated body parts, rather than just instance level.

region that is `most prominently camouflaged against the coral background`. This type of query probes the model's ability to discern the area of a camouflaged object that, paradoxically, remains the most visually salient or is least effectively concealed against its surroundings. Addressing such challenges correctly requires LVLMs to possess both fine-grained perceptual recognition and advanced reasoning skills pertinent to camouflage scenarios.

### B.4 Examples for IGS Task

Turning to the IGS task, Fig. 9 also illustrates examples incorporating both instance-level masks and our meticulously annotated part-level masks. In the left-hand example, which depicts two fish, the instruction given is `Segment the fish's head and fin on the right`. This command requires the model to perform spatial discrimination between the two fish instances and subsequently associate the designated region with a `[SEG]` token in its output. This token facilitates the downstream segmentation network in accurately delineating the specified parts. The example on the right demonstrates a part-level segmentation directive: `Segment the gecko's head`. Executing this instruction accurately demands that the LVLM precisely identifies specific anatomical regions of the target. This capability is particularly crucial in camouflaged environments where occlusions frequently occur and can obscure the complete form of the object.

### B.5 Examples for CEA Task

For the CEA task, Fig. 10 presents illustrative examples corresponding to four distinct levels of camouflage effectiveness: `Normal, Partial, Seamless, and Imperceptible`. Each example is accompanied by its assigned score and a textual explanation justifying the designated camouflage level. For the Camouflage Effectiveness Assessment (CEA) task, akin to the CAC task, we utilize a standardized prompt for querying all images, as depicted in the top-left corner of Fig. 10. As demonstrated in the figure, clear differentiations exist in the degree of camouflage across these levels.

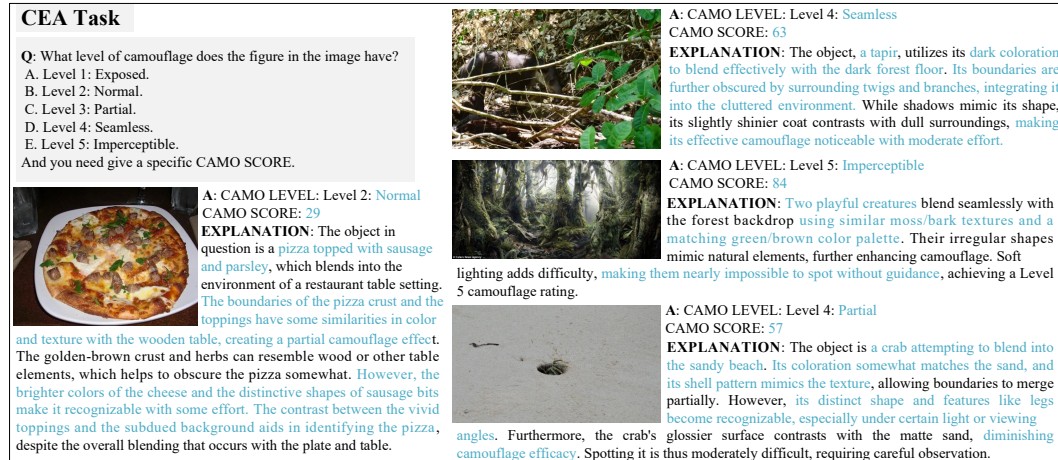

Figure 10: **Examples of the CEA task in MMCSBench**. In this figure, we show four different levels. It can be seen from the figure that the camouflage scores obtained by our algorithm can effectively represent the degree of camouflage of different targets.

Notably, at the `Normal` level, the presence and category of the target object are readily apparent, with its boundaries being sharply defined and easily discernible. As the camouflage level progresses to `Partial` and `Seamless`, the target's inconspicuousness increases. This includes instances where objects actively attempt concealment within their environment, such as a crab partially burying itself in the sand. Upon reaching the highest level, `Imperceptible`, the target becomes virtually impossible to detect visually.

Overall, the `Seamless` and `Imperceptible` levels represent instances of highly effective camouflage that are particularly significant for the development and evaluation of robust camouflage analysis systems. Consequently, our evaluation methodology places a specific emphasis on analyzing performance metrics, such as error rates, for these two challenging tiers.

### B.6 WordCloud for MMCSBench

Fig. 11 presents word cloud visualizations for the five tasks within MMCSBench across both the training and testing sets. Taking the CAC task as an example, we can clearly see keywords such as "camouflage", "surrounding", "color", "texture", "purpose", and "predator". This indicates that the CAC task focuses on describing object details, their surrounding environment, and camouflage strategies. The VQA task features more keywords related to object details, such as "color", "eye", and "body", suggesting that this task places greater emphasis on questions and answers regarding specific parts or details of the object, which highlights the fine-grained nature of our MMCSBench. Overall, the different tasks within MMCSBench have distinct focuses, allowing for the effective evaluation of existing LVLMs' performance in camouflage scenes.

## C CEA Task Supplement

### C.1 More CEA Level Visualization

In Fig. 12, we present images of different CEA levels to demonstrate the effectiveness of our camouflage score $C_s$ evaluation. As can be seen from the figure, as the camouflage level increases, the degree of camouflage and complexity of the images further rise. By calculating the camouflage score $C_s$, we can effectively assess the camouflage effectiveness of different object images, thereby training the LVLM.

### C.2 Landscape Visualization

In this section, we present a collection of 1473 landscape images that we have gathered in Fig. 13. These images span various types of scenery, such as seascapes, deserts, clouds, forests, and grasslands,

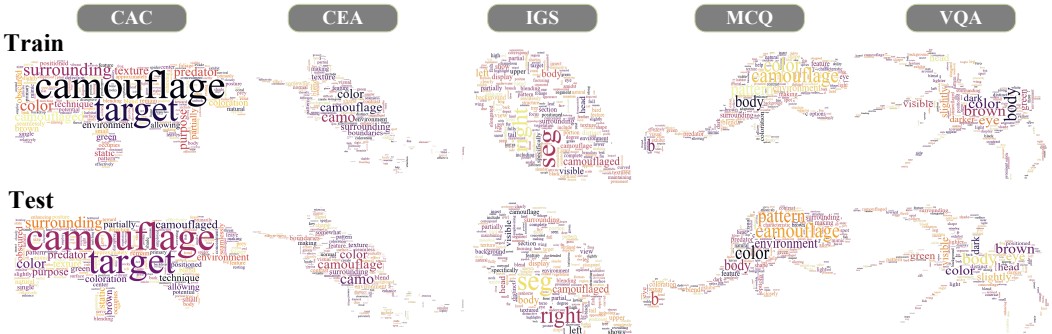

Figure 11: **Word cloud display of each task in MMCSBench**. For each task, the word cloud we generated only includes the answer part.

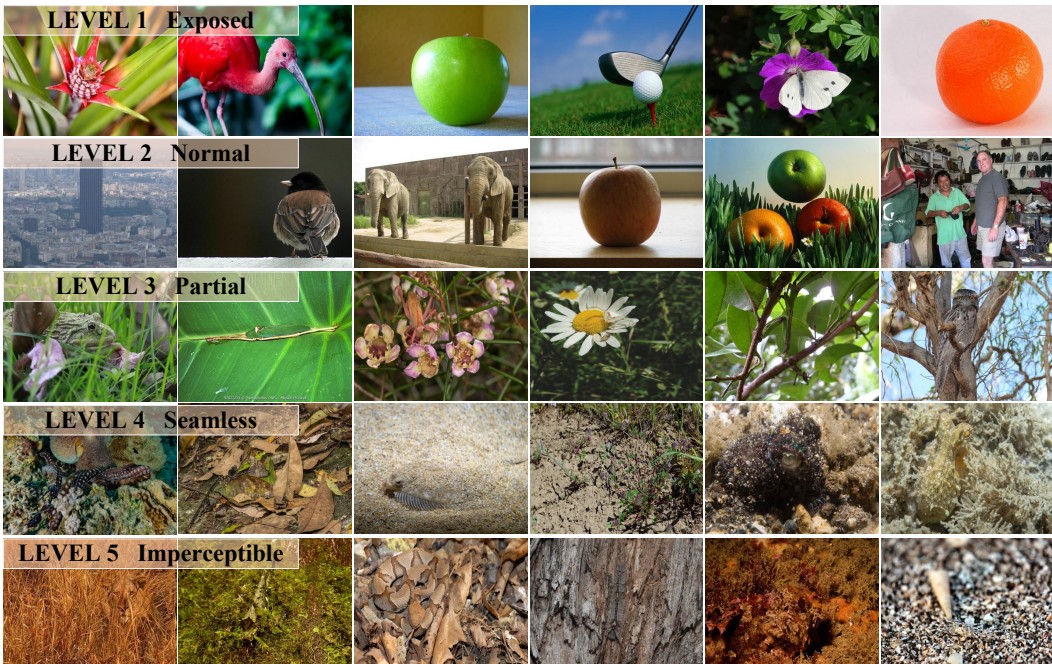

Figure 12: **Display of images at different CEA levels**. From top to bottom, the camouflage levels are 1-5. It can be seen from the figure that the lower the level, the simpler the overall image, while the higher the camouflage level, the more complex the background environment.

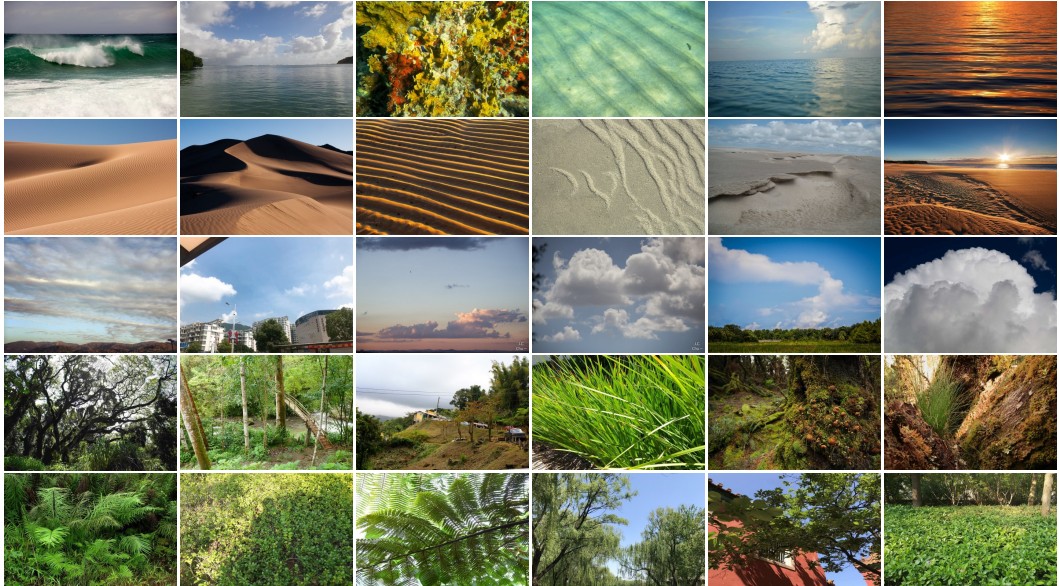

Figure 13: **Pure landscape image display**. We collected images from different perspectives such as sky, ocean, land, and grass, to broadly represent the living environments of various animals.

representing major habitats for different species. As can be observed from these images, they contain neither prominent objects nor camouflaged objects. Using these images for training CEA tasks can effectively prevent LVLMs from making the critical error of misinterpreting pure landscape images as containing highly camouflaged objects.

### C.3 50K+ Automatically Collected Camouflage Images

Fig. 14 demonstrates our automatic camouflage image detection and collection on the COCO [31], WildFish [73], iNat [56], and AK [44] datasets. The specific details are as follows:

1. We first fine-tuned Qwen2.5-VL-72B using 40% of the CEA data. Within this 40% subset, we aimed to maintain consistency in the number of samples across different camouflage levels. Additionally, we incorporated 1,473 collected pure landscape images into the training process.

2. We utilized the fine-tuned Qwen2.5-VL-72B to automatically assess the camouflage level of images within the COCO, WildFish, iNat, and AK datasets. We exclusively selected samples classified with the `Imperceptible` camouflage level.

3. For video stream data originating from the AK dataset, we calculated the Structural Similarity Index (SSIM) between images from the same video stream. To ensure sample diversity, images exhibiting an SSIM score greater than 0.5 were removed.

4. After de-duplication, we get the final 50K+ dataset.

Ultimately, this process resulted in the compilation of a dataset containing 54,468 camouflaged images, accompanied by their corresponding generated descriptions.

### C.4 Weight Analysis of $\alpha$, $\beta$ and $\gamma$ in $C_s$

To address the potential subjectivity in camouflage efficacy assessment and the varying importance of color, texture, and edge cues across different camouflage scenarios, we conducted a sensitivity analysis on the weights assigned to each component of the CEA score ($C_s$) in Table 6. Specifically, we explored the impact of different weight combinations ($\alpha$ for color similarity $S_c$, $\beta$ for texture similarity $S_t$, and $\gamma$ for edge fusion similarity $S_e$) on the overall $C_s$ and its correlation with LVLM performance. Our experiments further aimed to ensure that the distribution distinctions between camouflaged and natural images were consistently maintained across different weight settings. The

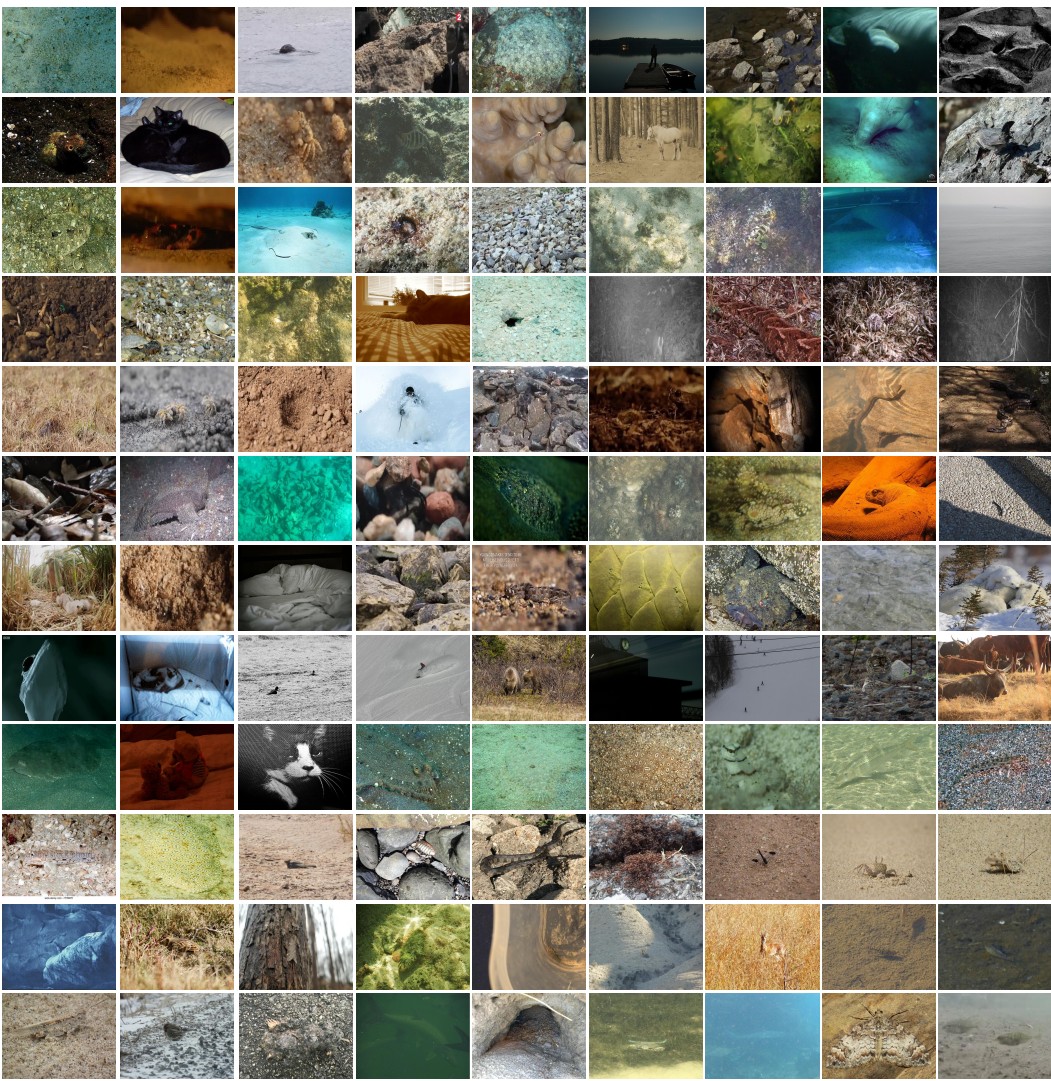

Figure 14: **Display of collected 50K+ camouflage images**. We fine-tuned Qwen2.5-VL-72B using the CEA task to enhance its understanding of camouflaged and non-camouflaged scenes. By incorporating pure landscape images, we enabled it to differentiate between different levels of camouflage. Subsequently, we automatically collected images from large-scale public datasets, and after deduplication, we formed a dataset of 50K+ camouflage images. During the filtering process, we only collected images that Qwen2.5-VL-72B assessed as being at camouflage level 5.

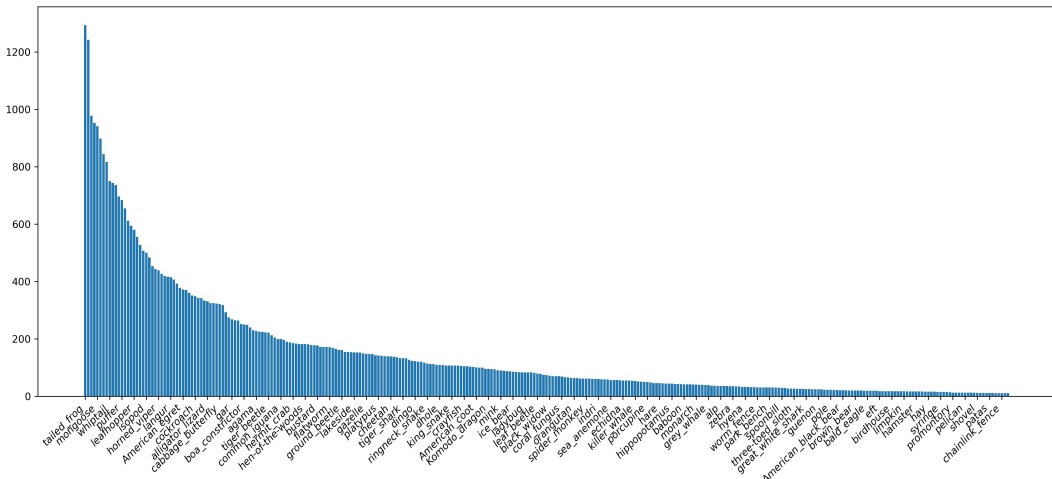

Figure 15: **Statistics of different categories in 50K+ camouflage images**. As you can see, similar to the previous dataset COD10K, the images we counted also showed a long-tailed distribution, because certain categories are naturally disguised, such as chameleons. Some categories were only extreme in certain cases.

results, detailed below, demonstrate that while individual $C_s$ values may shift slightly with weight adjustments, the relative ranking of camouflage effectiveness, the distribution separation, and the subsequent evaluation of LVLM performance remain largely consistent, thus justifying the robustness of our chosen weighting scheme. This approach aligns with methodologies used in prior work to establish reliable camouflage assessment metrics.

Table 6: **Sensitivity Analysis of CEA Score Weights**. $\alpha$, $\beta$, and $\gamma$ represent the weights for color similarity ($S_c$), texture similarity ($S_t$), and edge fusion similarity ($S_e$), respectively; $C_s$ denotes the overall CEA score; $\rho$ is the Spearman rank correlation coefficient.

| Weight Scheme | Weights | | | Avg. $C_s$ | $C_s$ Std Dev | $\rho$ with Baseline | Qwen2.5-VL-7B | |
|---|---|---|---|---|---|---|---|---|
| | $\alpha$ | $\beta$ | $\gamma$ | | | | CEA Acc | CEA MAE |
| Baseline | 0.5 | 0.3 | 0.2 | 49.01 | 21.17 | 1.00 | 72.84 | 6.32 |
| Color | 0.7 | 0.15 | 0.15 | 49.52 | 20.56 | 0.93 | 71.50 | 6.47 |
| Texture | 0.15 | 0.7 | 0.15 | 51.74 | 18.87 | 0.87 | 72.65 | 6.25 |
| Edge | 0.15 | 0.15 | 0.7 | 42.93 | 19.62 | 0.86 | 72.72 | 6.30 |
| Equal | 1/3 | 1/3 | 1/3 | 49.21 | 21.30 | 0.91 | 71.75 | 6.85 |

# D    CamoVLM for Camouflage Scene Understanding

Previous approaches, such as LISA [24] and GeoPixel [2], commonly employ the Segment Anything Model (SAM) as their segmentation network. However, SAM encounters difficulties with fine-grained segmentation in camouflage scenes. It often necessitates high-resolution inputs and multiple iterations for mask refinement, leading to significant resource consumption. More critically, SAM lacks an effective mechanism for multimodal alignment, which can result in incorrect instruction comprehension and the misidentification of non-target regions.

To address these limitations, we introduce CamoVLM, a large vision-language model specifically designed for camouflage scene understanding, as illustrated in Fig. 16. CamoVLM adheres to the established framework of grounding LVLMs and comprises two key components: (1) An LVLM branch, utilizing Qwen2.5-VL-7B [59], dedicated to tasks such as CAC, VQA, CEA and MCQ. (2) A grounding segmentation branch, CamoPixelModel, engineered for precise object segmentation. Our segmentation branch builds upon existing COD models and incorporates a proposed lookup matching strategy to enhance the alignment between visual features and segmentation tokens. This method generates an attention mask tailored to the prompted region, enabling the decoder to follow

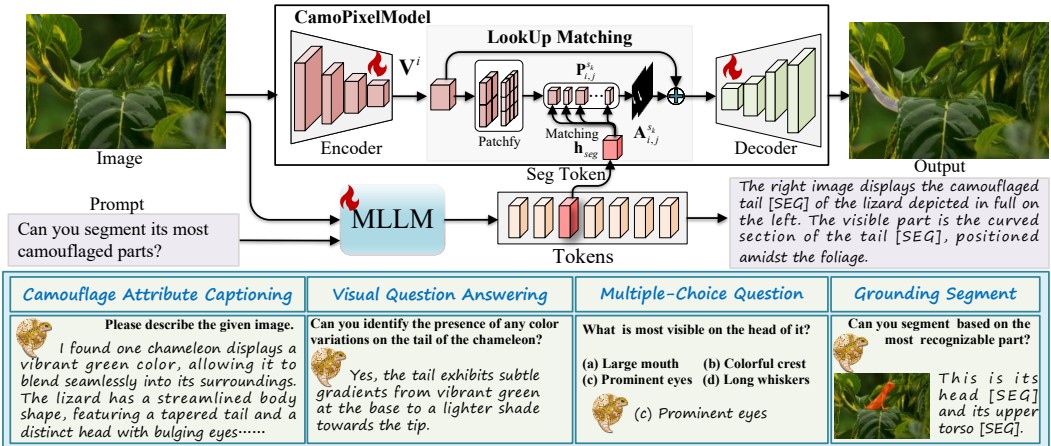

Figure 16: **Overall framework diagram of the proposed CamoVLM**. CamoVLM includes an LVLM branch and a segmentation branch. The LVLM branch passes the token obtained by the [SEG] identifier to the segmentation branch, thereby performing instruction segmentation and IGS tasks. Simultaneously, the LVLM branch can handle CAC, VQA, MCQ, and CEA tasks.

instructions with greater precision. The subsequent sections will provide a detailed explanation of both the LVLM and segmentation branches.

## D.1 LVLM and [SEG] Token

For tasks such as CAC, VQA, CEA and MCQ, we can conveniently train them with existing LVLMs. However, for IGS tasks, a special token is required to guide the LVLM in generating tokens specifically for segmentation.

Inspired by LISA's [24] paradigm of injecting segmentation capabilities into LVLMs, CamoVLM adopts a similar mechanism to enable joint reasoning and segmentation for camouflage scene understanding. The core idea involves integrating a special token [SEG] into the LVLM's vocabulary to trigger segmentation mask generation while preserving the model's language comprehension and reasoning abilities.

Given an input visual image $\mathbf{x}_v$ and a text instruction $\mathbf{x}_t$, the LVLM branch (Qwen2.5-VL-7B) processes them to generate a textual response $\hat{\mathbf{y}}_t$. When the instruction requires segmentation (e.g., "Segment the camouflaged insect on the tree"), the model outputs the [SEG] token behind the token "insect". Formally:

$$\hat{\mathbf{y}}_t = \mathcal{F}_{LVLM}(\mathbf{x}_v, \mathbf{x}_t), \tag{4}$$

where $\mathcal{F}_{LVLM}$ represents the LVLM. The hidden embedding $\tilde{\mathbf{h}}_{seg}$ corresponding to the [SEG] token is extracted from the final layer of the LVLM. This embedding is then projected into a segmentation-aware feature $\mathbf{h}_{seg}$ via a lightweight MLP layer $\gamma$:

$$\mathbf{h}_{seg} = \gamma(\tilde{\mathbf{h}}_{seg}). \tag{5}$$

## D.2 Grounding Segmentation and LUM

**CamoPixelModel**. To effectively address the cross-modal mismatch challenges inherent in camouflage scenes, we developed CamoPixelModel to serve as the segmentation branch of CamoVLM. CamoPixelModel adopts a U-Net-like architecture [51], comprising an encoder, our proposed matching strategy, and a decoder. It processes both the input image and the segmentation token.

Given an input image $\mathbf{x}$, CamoPixelModel employs ConvNeXt [38] as its encoder ($\mathcal{F}_E$) to extract hierarchical feature representations:

$$\mathbf{V}^m = \mathcal{F}_E(\mathbf{x}), \quad m \in \{1, 2, 3, 4\}, \tag{6}$$

where $\mathcal{F}_E$ captures multi-scale visual features $\mathbf{V}^m$. These extracted features subsequently undergo processing via the LookUp Matching strategy (detailed below), which aligns them with segmentation

tokens, thereby enhancing region-specific understanding. Finally, the decoder integrates the [SEG] token embeddings, facilitating precise mask generation and improving adherence to instructions. This structured approach enables CamoPixelModel to achieve superior pixel-level segmentation accuracy compared to models like SAM, while concurrently maintaining computational efficiency.

**LookUp Matching Strategy**. Camouflaged objects inherently blend into their surroundings due to similarities in texture, color, and pattern. This poses a significant challenge for achieving precise feature alignment between textual instructions and visual content. To overcome this fundamental difficulty, we propose the LUM strategy, a specialized cross-modal alignment mechanism tailored for camouflage scenarios.

Specifically, LUM operates by processing visual features at multiple patch scales. This multi-scale approach assists the model in locating objects, as specified by the prompt, within the relevant feature regions. Given the visual features $\mathbf{V}^m$ and the segmentation token embedding $\mathbf{h}_{\text{seg}}$, the module extracts patches at various scales:

$$\mathbf{P}_{i,j}^{s_k} = \Phi\left(\mathcal{U}_{s_k}(\mathbf{V}^m)_{i,j}\right) \in \mathbb{R}^{d \times s_k \times s_k}, \quad s_k \in \mathcal{S} = \{s_1, s_2, \ldots, s_n\}, \tag{7}$$

where $\mathbf{P}_{i,j}^{s_k}$ denotes the patch at position $(i, j)$ with size $s_k$. The operation $\mathcal{U}_{s_k}$ unfolds the feature map $\mathbf{V}^m$ to extract $s_k \times s_k$ patches. $\Phi$ represents a projection function mapping the patch into the embedding space, and $\mathcal{S}$ defines the set of configurable patch sizes, enabling multi-scale feature alignment.

Subsequently, to perform the lookup matching between patches $\mathbf{P}_{i,j}^{s_k}$ and the embedding $\mathbf{h}_{\text{seg}}$, a cross-attention mechanism is applied to compute scores for each patch:

$$\mathbf{A}_{i,j}^{s_k} = \text{softmax}\left(\frac{\mathbf{P}_{i,j}^{s_k} \cdot \mathbf{h}_{\text{seg}}^{\top}}{\sqrt{d}}\right) \cdot \mathbf{h}_{\text{seg}}, \tag{8}$$

where $\mathbf{A}_{i,j}^{s_k}$ signifies the cross-attention output for patch $(i, j)$ at scale $s_k$. This step calculates attention scores between visual patches and segmentation tokens via scaled dot-product attention.

Finally, the enhanced feature map $\mathbf{V}_{\phi}^m$ is produced by integrating the multi-scale attention results with the original features:

$$\mathbf{V}_{\phi}^m = \psi\left(\mathcal{G}\left(\mathbf{V}^i \oplus \bigcup_{k=1}^{n} \mathcal{R}(\mathbf{A}^{s_k})\right)\right), \tag{9}$$

where $\oplus$ indicates channel-wise concatenation, $\mathcal{R}$ is a reshaping operation that restores the spatial dimensions of the attention maps, $\mathcal{G}$ denotes a convolutional compression layer designed to fuse the attention-enhanced features, and $\psi$ represents the ReLU activation function. This integration of multi-scale patch information empowers the model to effectively locate camouflaged regions across various levels of granularity.

**Decoder**. After applying the LUM strategy, we obtain text-aligned hierarchical features $\mathbf{V}_{\phi}^m$. Subsequently, a hierarchical decoder is employed to perform progressive fusion of multi-scale features from deep to shallow layers. For detailed decoder architecture, please refer to [68]. The decoded features ultimately generate the output based on text prompts, as visualized in Fig. 16.

# E  Implementation Details of Training and Testing

## E.1  Implementation Details

In our study, all Large Vision-Language Models (LVLMs) are fine-tuned using the Low-Rank Adaptation (LoRA) methodology [20]. Specifically, the LoRA rank ($r$) is set to 8, and the scaling factor $\alpha$ is configured to 16. Throughout the fine-tuning process, both the visual encoder and the LLM backbone remain unfrozen, allowing their parameters to be updated. The batch size is dynamically adjusted between 1 and 4, contingent upon the parameter size of the specific model, to optimize GPU memory utilization. All models are trained for a single epoch. We employ the AdamW [40] optimizer, and the learning rate is scheduled using a cosine decay strategy, annealing from an initial value of $1 \times 10^{-4}$ down to 0.

Regarding data selection, we construct a training subset corresponding to 1% of the total available data, as shown in Table 7. For the CEA task, we ensure an equal distribution of samples across its

five distinct levels within this subset. This CEA-specific data is then augmented with an equivalent volume of pure scenic images, along with text-image pairs sourced from other auxiliary tasks. For the IGS task, we utilize 50% of its available data for training, as preliminary experiments indicate that using only 1% is insufficient for adequate model convergence. All experiments are conducted on a system equipped with 8 NVIDIA A6000 GPUs, accommodating LVLMs of varying scales.

When training text generation tasks, CAC, VQA, MCQ and CEA, we use the cross-entropy loss function. When training IGS tasks, we use both the binary cross-entropy loss function and the IoU loss function. By default, the weight of each loss function is 1.

Table 7: **Number of Training Samples for Different Tasks in MMCSBench**.

| Task | Number of Training Samples |
|------|----------------------------|
| CAC  | 125  |
| VQA  | 250  |
| MCQ  | 232  |
| CEA  | 210  |
| IGS  | 1523 |

### E.2  Testing Details for No fine-tuned LVLMs

When directly querying models such as GPT-4.1, Gemini 2.5-Pro, and Qwen2.5-VL-72B with camouflage-related questions, their responses often lack the standardized format consistent with our desired output. Therefore, we introduce specific prompts to constrain their output, compelling them to respond in a standardized manner, which allows for a more authentic validation of their understanding in camouflage scenarios.

Specifically, for the CAC task, we append the prompt `Please describe in one paragraph the quantity, category, size, position, color, camouflage strategy, and camouflage purpose.` to the question `How is the camouflaged target integrated into its environment?`. For the MCQ task, after the LVLMs provide their answers, we also prompt them with `Please provide a paragraph explaining your choice.` to elicit justifications. For the CEA task, we similarly require them to provide a specific score, constrained to be within 100, informing the LVLMs that a higher score indicates a higher degree of camouflage.

It is important to note that this prompting strategy is specifically applied to non-fine-tuned LVLMs to encourage them to generate standardized outputs, which are subsequently used for metric calculation. Fine-tuned LVLMs, having adapted to the standardized output format during their training, do not require such elaborate prompting.

### E.3  Training Performance on Different Tasks

Fig. 17 shows the training performance of our MMCSBench. Specifically, for a given image, we present the Qwen2.5-VL-7B outputs alongside the ground truth texts for various tasks.

### E.4  Training Performance with Different Number of Samples

To investigate the impact of training data size on LVLM performance within camouflage scenes, we conducted experiments using Qwen2.5-VL-7B with varying percentages of the MMCSBench dataset. We evaluated performance on the CAC, VQA, MCQ, and CEA tasks using 2%, 5%, 10%, 20%, and 50% of the training data, comparing the results to the baseline 1% training scenario, as shown in Table 8. Our findings indicate that increasing the training data generally improves performance, but with diminishing returns observed beyond 10% or 20%, where performance plateaus. This saturation effect can be attributed to several factors, including potential redundancy within the dataset itself, limitations in the model's capacity to extract increasingly nuanced features from larger data volumes, the inherent complexity of the camouflage understanding task, and the constraints imposed by the fine-tuning methodology. Furthermore, we observed that the overall distribution of camouflage scores remained relatively consistent across different training set sizes, suggesting that while the

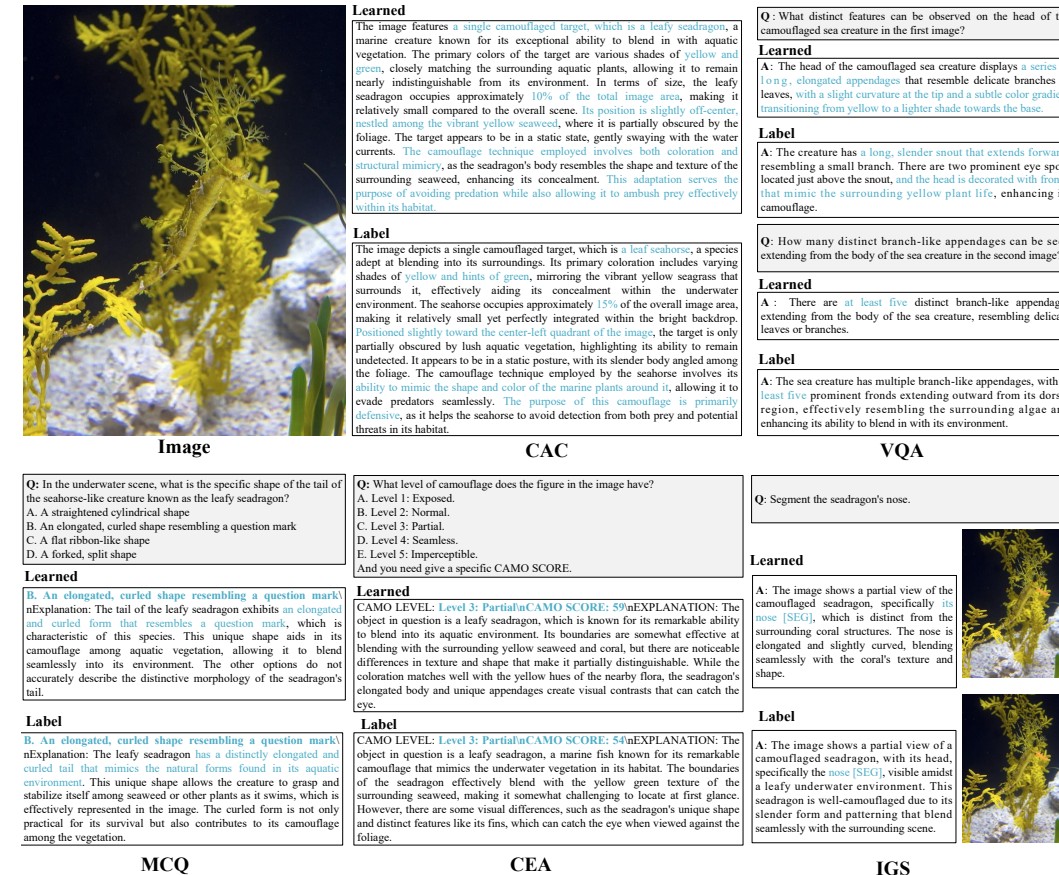

Figure 17: **The performance of Qwen2.5-VL-7B on five tasks using the same camouflaged image**. We used one image to test the performance of Qwen2.5-VL-7B on CAC, VQA, MCQ, CEA, and IGS tasks. In the figure, we present "Learned" to denote the output of Qwen2.5-VL-7B and "Label" to denote our annotated ground truth text. It can be observed that Qwen2.5-VL-7B exhibits good performance across different tasks. For example, in CAC, it can accurately predict the category, quantity, position, and camouflage strategy, even though there are errors in size estimation. In VQA and MCQ tasks, it can answer correctly. Notably, in the CEA task, it can accurately judge the degree of camouflage and the score (59 vs. 54), and in the IGS task, it also demonstrates accurate segmentation.

model's ability to precisely predict scores may improve, its overall understanding of camouflage levels stabilizes early in the training process.

Table 8: **Qwen2.5-VL-7B Performance with Varying Training Data Percentage**. There may be many reasons for the change in metrics, including a reduction in the amount of data in the test set.

| Train Data % | CAC | | | VQA | | | MCQ | | | CEA | | |
|---|---|---|---|---|---|---|---|---|---|---|---|---|
| | BLEU | METEOR | RougeL | BLEU | METEOR | RougeL | ACC | BLEU | METEOR | ACC | MAE | BLEU |
| 1% (Baseline) | 18.09 | 42.52 | 33.87 | 19.25 | 45.10 | 41.83 | 75.20 | 25.71 | 43.61 | 72.84 | 6.32 | 20.25 |
| 2% | 18.71 | 43.15 | 34.45 | 19.95 | 45.68 | 42.28 | 76.15 | 26.12 | 44.18 | 73.62 | 6.18 | 20.68 |
| 5% | 19.28 | 43.92 | 35.21 | 20.62 | 46.29 | 43.17 | 77.23 | 26.67 | 44.91 | 74.19 | 5.97 | 21.14 |
| 10% | 19.45 | 44.28 | 35.68 | 20.91 | 46.52 | 43.42 | 77.78 | 26.94 | 45.24 | 74.43 | 5.85 | 21.31 |
| 20% | 19.51 | 44.35 | 35.75 | 20.98 | 46.59 | 43.49 | 77.82 | 26.98 | 45.29 | 74.48 | 5.81 | 21.38 |
| 50% | 19.53 | 44.38 | 35.78 | 21.01 | 46.62 | 43.52 | 77.85 | 27.01 | 45.32 | 74.51 | 5.78 | 21.41 |

