# OpenReview forum: "MMCSBench: A Fine-Grained Benchmark for Large Vision-Language Models in Camouflage Scenes"
_NeurIPS.cc/2025/Datasets_and_Benchmarks_Track — NeurIPS 2025 Datasets and Benchmarks Track poster_

### Official Review · Reviewer_cgFu · 2025-07-01

**Ethics Flags:** Safety and security
**Rating:** 5
**Confidence:** 5

**Summary:**

This paper introduces MMCSBench, a large, fine-grained multimodal benchmark specifically designed to assess and improve the capabilities of large vision-language models (LVLMs) on challenging camouflage scenes. The benchmark comprises 22,537 images and 76,843 image-text pairs spanning five distinct tasks, including Camouflage Attribute Captioning, fine-grained Visual Question Answering, Multiple-Choice Questions, Image Grounding Segmentation, and a novel Camouflage Efficacy Assessment (CEA) task. The dataset is meticulously annotated with both pixel-level and textual labels. Extensive empirical studies are conducted on 26 state-of-the-art LVLMs, revealing their significant limitations in camouflaged environments and providing a resource and pathway for future research.

**Dataset Code Accessibility:**

Yes

**Ethical Considerations:**

No, there are no or only very minor ethics concerns

**Final Justification:**

All concerns are addressed after rebuttal, and I decide to accept this paper.

**Limitations Weaknesses:**

1. While the dataset covers a broad range of camouflage types and objects, the primary focus remains on non-human, animal, and natural scenes; less attention is given to artificial or adversarial camouflage or broader cross-domain generalization.

2. The CEA scoring, though based on clear metrics and manual review, relies on specific feature choices and weighted sums. Alternative definitions or weightings could influence scores. The paper acknowledges this (Limitations), but empirical validation against diverse annotator populations or downstream tasks could strengthen confidence.

3. Table 2 and Table 3 provide extensive benchmarking, but some newer or specialized camouflage detection models (outside of large multimodal VL family) are not compared, and more discussion of why certain baselines were omitted would strengthen claims

**Strengths Contributions:**

1. Comprehensive Benchmark: MMCSBench is the first large-scale, multi-task benchmark focusing on camouflage scenes for LVLMs. It introduces a carefully constructed and well-justified dataset covering multiple modalities (image-text pairs, pixel-level masks) and tasks.

2. Novel Task of Camouflage Efficacy Assessment (CEA): The proposal and formalization of the CEA task addresses a previously unexplored but practically relevant challenge: quantifying camouflage effectiveness, which also enables automated discovery and annotation of camouflaged objects in large datasets.

3. The dataset spans 22,537 images, 76,843 image-text pairs, and multiple public sources, addressing data scarcity in camouflage detection.

4. The experiments cover a wide range of state-of-the-art LVLMs, including zero-shot and fine-tuned settings, with clear comparisons to human performance and careful protocol.

---

> ### Author Rebuttal · Authors · 2025-07-25
>
> Thank you very much for your thorough review and understanding of our work. Much like **Reviewer knoJ**, you have provided extremely valuable suggestions, and we sincerely appreciate your positive feedback. We will now address your questions and comments one by one. Thank you again!
>
> ## **1. While the dataset covers a broad range of camouflage types and objects, the primary focus remains on non-human, animal, and natural scenes; less attention is given to artificial or adversarial camouflage or broader cross-domain generalization.**
>
> Thank you for this insightful suggestion. This is a very constructive point that could only come from a detailed reading of our work, and we sincerely appreciate it.
>
> Regarding images of artificial or adversarial camouflage, you are correct that this is an important area. Our MMCSBench does contain a subset of such images from CAMO [1] and COD10K [2] datasets. We have analyzed this, and **they account for approximately 825 images**, or about **8% of the total camouflage images**. Such images include **soldiers wearing** camouflage uniforms or **body painting** adversarial camouflage that blends in with nature. We acknowledge that this category was not well represented in our manuscript, which may have led to the reviewers’ misjudgment of the diversity of our dataset.
>
> Your feedback has underscored the importance of this area, and we are committed to diversifying our dataset in the future. This will include actively collecting more varied data, including from domains like medical imaging and even synthetic data. **Thank you again for your valuable suggestion; it is extremely helpful for the improvement of our work.**
>
> ## **2. The CEA scoring, though based on clear metrics and manual review, relies on specific feature choices and weighted sums. Alternative definitions or weightings could influence scores. The paper acknowledges this (Limitations), but empirical validation against diverse annotator populations or downstream tasks could strengthen confidence.**
>
> Thank you for this very insightful and important point. We completely agree that the robustness and validity of the CEA scoring mechanism are critical to our benchmark's contribution.
>
> To address the potential subjectivity of the feature weights, not only did we ground our approach in established metrics from prior work, but to empirically validate our chosen weights, we also conducted a detailed **sensitivity analysis**. The full results are presented in **Appendix C.4 and Table 6**. The key finding from this analysis is that the relative ranking of camouflage effectiveness, as well as the downstream model performance, **remained largely consistent and robust** even across different weighting schemes. This gave us confidence in the reliability of our current configuration.
>
> That being said, we believe you have raised an excellent point regarding further validation against diverse human annotator populations. While our current work has established the metric's computational robustness, a large-scale study correlating the CEA score with human perceptual judgments is indeed a valuable and exciting direction for future research.
>
> Thank you again for your constructive feedback, which has helped us clarify our current validation and has provided valuable inspiration for our future work agenda.
>
> ## **3. Table 2 and Table 3 provide extensive benchmarking, but some newer or specialized camouflage detection models (outside of large multimodal VL family) are not compared, and more discussion of why certain baselines were omitted would strengthen claims**
>
> Thank you for this thoughtful question regarding our choice of baseline models. You have raised an important point about the scope of our comparisons. Our primary goal with MMCSBench is to introduce the first comprehensive benchmark specifically designed to evaluate and advance the **fine-grained, multimodal understanding capabilities of LVLMs** in camouflage scenes. This represents a shift in focus from traditional COD, which we see as a discriminative segmentation task, towards a more holistic, human-like reasoning about *why* and *how* an object is camouflaged.
>
> This core objective is directly reflected in our choice of baselines and explains the omission of most traditional, specialized COD models:
>
>  **Focus on Instruction-Based Segmentation (IGS):** Even for our segmentation task, IGS (Table 3), a direct comparison with traditional COD models is not appropriate. The IGS task is designed to evaluate **grounding segmentation**, where a model must interpret a natural language prompt (e.g., "Segment the most camouflaged part of the spider") to perform the segmentation. Traditional COD models lack this instruction-following ability. Therefore, we deliberately selected state-of-the-art *grounding segmentation LVLMs* like LISA, GLAMM, and GeoPixel as the most relevant baselines for this specific task.
> ***
>
> That being said, we recognize the value in understanding how the segmentation component of our CamoVLM, **CamoPixelModel**, performs in a traditional, single-modality COD setting. To provide this direct comparison and situate its performance, we conducted an additional experiment benchmarking CamoPixelModel against recent state-of-the-art COD models, **RUNNet [4]** and **VScode [5]**, on the popular **COD10K test set**. All models were trained on the official COD10K training set.
>
> The results, presented in Table I, are evaluated using standard COD metrics (S-measure $S_{\alpha}$, E-measure $E_{\phi}$, mIoU, and MAE).
>
> **Table I: Performance comparison on the COD10K test set for single-modality camouflaged object detection.**
>
> | Model | $S_{\alpha}$↑ | $E_{\phi}$↑ | $F_{\beta}$↑ | MAE↓ |
> | :--- | :---: | :---: | :---: | :---: |
> | VScode | 0.847 | 0.925 | **0.795** | 0.028 |
> | RUNNet | 0.857 | 0.926 | 0.769 | 0.027 |
> | **CamoPixelModel (Ours)** | **0.862** | **0.929** | 0.773 | **0.026** |
>
> As the results show, our CamoPixelModel demonstrates a leading performance by outperforming the baselines on the majority of key metrics ($S_{\alpha}$, $E_{\phi}$, and MAE). Although VScode shows a strong result on the $F_{\beta}$ metric, our model's overall capability is highly competitive. This strong performance in a pure segmentation task validates our architectural choices and demonstrates that its design for multimodal alignment does not come at the cost of segmentation quality. It confirms that CamoPixelModel serves as a powerful and effective segmentation foundation for the full CamoVLM framework.
>
> In summary, while our main benchmark focuses on the unique, multimodal reasoning capabilities of LVLMs, we have also demonstrated the state-of-the-art performance of our segmentation module in a traditional context. We recognize that clarifying this distinction is crucial. In our revision, we will add this comparative analysis and make the rationale for our baseline selections even more explicit. **Thank you for helping us identify this opportunity to strengthen the paper's clarity and rigor.**
>
> ***We sincerely thank you for your recognition and encouragement of our work. Your comments clearly reflect a meticulous review of our paper and, just as importantly, a shared vision to promote the development of the camouflage field, just as we hope to do. In a new version of the paper, we will further discuss disguise in other areas and give a clearer explanation of CEA's original purpose. We are deeply grateful for your invaluable suggestions!!!***
>
> [1] Le T N, Nguyen T V, Nie Z, et al. Anabranch network for camouflaged object segmentation[J]. Computer vision and image understanding, 2019, 184: 45-56.
>
> [2] Fan D P, Ji G P, Sun G, et al. Camouflaged object detection[C]//Proceedings of the IEEE/CVF conference on computer vision and pattern recognition. 2020: 2777-2787.
>
> [3] Luo Z, Liu N, Zhao W, et al. Vscode: General visual salient and camouflaged object detection with 2d prompt learning[C]//Proceedings of the IEEE/CVF conference on computer vision and pattern recognition. 2024: 17169-17180.
>
> [4] He C, Zhang R, Xiao F, et al. Run: Reversible unfolding network for concealed object segmentation[J]. arXiv preprint arXiv:2501.18783, 2025.

---

> > ### Comment · Reviewer_cgFu · 2025-08-05
> > **Thanks for rebuttal**
> >
> > Thank you for rebuttal! All concerns are addressed, and I decide to accept this paper.

---

> > > ### Author Response · Authors · 2025-08-05
> > > **Sincerely thank you for your advice and support of our work!**
> > >
> > > **Thank you for your positive feedback on our work. We'll incorporate your suggestions into the final version. We sincerely appreciate your contributions to both our work and the field of camouflage.**

---

### Official Review · Reviewer_knoJ · 2025-07-02

**Rating:** 5
**Confidence:** 3

**Summary:**

This paper presents MMCSBench, the first comprehensive multimodal benchmark specifically designed to evaluate and improve the performance of Large Vision-Language Models (LVLMs) in camouflage scene understanding. The benchmark includes 22,537 images and 76,843 image-text pairs across five fine-grained tasks. The authors also propose CamoVLM, a specialized LVLM for camouflage understanding and grounding. Extensive experiments across 26 LVLMs (ranging from 2B to 72B parameters) show that current models struggle significantly with camouflage scenarios. The CEA task further enables automated collection of camouflaged imagery from large-scale datasets.

**Additional Feedback:**

This work presents MMCSBench, a novel multimodal benchmark specifically designed to assess the capabilities of Large Vision-Language Models (LVLMs) in camouflaged object detection and understanding. The benchmark consists of 22,537 carefully annotated images paired with 76,843 textual descriptions, organized into five distinct evaluation tasks. The authors propose CamoVLM, a baseline model for camouflage scene understanding, and conduct extensive experiments on 26 LVLMs, revealing significant limitations in their ability to interpret camouflaged objects. A key innovation is the proposed CEA task, which enables automated scoring of camouflage effectiveness and facilitates large-scale collection of additional camouflage images.

Here are some areas of concern:

1.From my point of view, it would be useful to consider evaluating models on various datasets like completely unseen camouflage datasets (e.g., synthetic data) to assess broader applicability of VLMs.

2.Despite fine-tuning, even the strongest LVLMs (e.g., Qwen2.5-VL-72B, InternVL3-78B) show substantial performance gaps compared to human annotators. This suggests that the current model architectures or training strategies may fundamentally lack mechanisms to handle camouflage reasoning. So it would be helpful if the paper discussed why current model architectures struggle—e.g., is it due to visual-textual misalignment, attention dispersion in dense scenes, or insufficient visual grounding? Including this reflection may guide future model design.

3.The paper presents extensive quantitative results, but a qualitative analysis of failure cases—such as common misinterpretations in VQA, poor segmentation in IGS, or hallucinated descriptions in CEA—would provide valuable insights. A few visual examples with model predictions and human references could strengthen the reader’s understanding of current limitations.

Overall:
This is a high-quality, well-executed paper that makes a significant contribution to multimodal AI research. The benchmark is novel, well-designed, and empirically validated, providing a strong foundation for future work.

**Dataset Code Accessibility:**

Yes

**Ethical Considerations:**

No, there are no or only very minor ethics concerns

**Final Justification:**

I am generally satisfied with the reply.  But I will keep my score because of the first concerns.

**Limitations Weaknesses:**

1.Limited generalization analysis
2.Limited performance improvement of sota LVLMs
3.Limited analysis of failure cases

**Strengths Contributions:**

1.Trendy and interesting topic
2.Novel benchmark design
3.Comprehensive analysis and tasks design
4.Well-written and well-structured paper

---

> ### Author Rebuttal · Authors · 2025-07-25
>
> We sincerely appreciate your recognition of our work. It is evident from your review that you have read our work thoroughly, and for that, we are very grateful. Thank you for your diligent review. We will now address your comments and questions individually. We would like to once again thank you for your generous and selfless review.
>
> ## **1. Evaluating on more various datasets like completely unseen camouflage datasets (e.g., synthetic data).**
>
> That is a very constructive suggestion. We did indeed consider the richness and diversity of the dataset. As we mention in the paper, **resources for camouflaged object datasets remain quite scarce**. The only ones that can be effectively utilized are the existing public datasets, such as CAMO [1], COD10K [2], and CHAMELEON [3]. This scarcity presented a significant **challenge for the construction of our MMCSBench**. We attempted numerous methods to generate synthetic camouflage data, for instance, using **LAKE-RED [4]** and **Stable Diffusion [5]**, but these efforts were unsuccessful. The data produced by these methods failed to achieve a convincing degree of camouflage, compelling us to use only the existing COD dataset.
>
> It is precisely this limitation that prompted us to initiate the **CEA task**. We believe this is an effective method for improving the resources available to the camouflage community. In our work, we have demonstrated the effectiveness of CEA. Through this task, we have successfully collected 50,000 new camouflaged images along with their corresponding text descriptions. We are committed to keeping the project open-source to foster community development. We hope that **the MMCSBench and the CEA task will stimulate future work, such as the development of diffusion models specifically for camouflage generation and the advancement of research into LVLMs focused on camouflage.**
>
> ## **2. The paper would be enhanced by a discussion on the root causes—such as visual-textual misalignment or attention dispersion—for the significant performance gap in camouflage reasoning observed even in powerful, fine-tuned LVLMs.**
>
> That is another highly constructive suggestion. Thank you once again for your meticulous review and for recognizing the importance of this issue. We would be delighted to elaborate on our perspective regarding why existing LVLMs tend to fail on camouflaged images.
>
> Our hypothesis is that this issue is fundamentally similar to the challenges observed when earlier models, like the Segment Anything Model (SAM), performed poorly on tasks such as Camouflaged Object Detection (COD). The core problem, we believe, lies with the **visual encoder**.
>
> Conventional visual encoders are generally trained to apply attention to salient, large-scale objects within an image. However, camouflaged targets are, by their nature, cryptic and small. Consequently, when an LVLM processes a camouflage image, its visual encoder may prioritize and pass tokens representing the background or other prominent objects, rather than the intended camouflaged target. This issue is potentially exacerbated in architectures like Qwen-VL, which employ **visual token compression**, leading to a further loss of the subtle information that defines the target. This explains why Qwen-VL consistently underperforms compared to other models on the CAC task, as shown in Table 2 of the manuscript. When the LLM component receives these visual tokens that are **misaligned with the user's prompt** (e.g., "find the hidden spider"), a conflict arises, making it highly probable that the model will produce a **hallucination** or an otherwise incorrect answer.
>
> In summary, we attribute the failure of LVLMs to a **misalignment between the visual encoder and the text**: the encoder's **inherent saliency bias** makes it ill-suited for identifying non-salient camouflaged objects, while mechanisms like **token compression** can further erase the critical details required for successful recognition. We thank you for prompting us to reflect on this, as it is vital for guiding future model design.
>
> **We will include this discussion in the manuscript. Thank you for your suggestion.**
>
> ## **3.  A few visual examples with model predictions and human references could strengthen the reader’s understanding of current limitations.**
>
> Thank you for this excellent and highly constructive suggestion. We completely agree that a qualitative analysis of failure cases would provide valuable insights and significantly strengthen the paper.
>
> In the revised version of our manuscript, **we will add a new subsection dedicated to this qualitative analysis**. Following your recommendation, this section will feature several visual examples from key tasks like VQA, IGS, and CEA. Each example will present a side-by-side comparison of:
> -  The input image.
> -  The model's erroneous output (e.g., a misinterpreted answer, poor segmentation, or a hallucinated description).
> -  The human reference (ground truth).
>
> Furthermore, we will accompany these visuals with a brief discussion on the nature of the failure to deepen the reader's understanding of the current limitations. We are confident that these additions will greatly enhance the paper's clarity and impact. Thank you again for this valuable feedback.
>
> ***Above the response, additional analysis will be incorporated in the next version, including a discussion on the scope of datasets in the related work and an analysis of our model's limitations in the experimental section. Thank you very much for your understanding of our work and thank you again for your careful review of our paper!!!***
>
> [1] Le T N, Nguyen T V, Nie Z, et al. Anabranch network for camouflaged object segmentation[J]. Computer vision and image understanding, 2019, 184: 45-56.
>
> [2] Fan D P, Ji G P, Sun G, et al. Camouflaged object detection[C]//Proceedings of the IEEE/CVF conference on computer vision and pattern recognition. 2020: 2777-2787.
>
> [3] Skurowski P, Abdulameer H, Błaszczyk J, et al. Animal camouflage analysis: Chameleon database[J]. Unpublished manuscript, 2018, 2(6): 7.
>
> [4] Zhao P, Xu P, Qin P, et al. Lake-red: Camouflaged images generation by latent background knowledge retrieval-augmented diffusion[C]//Proceedings of the IEEE/CVF Conference on Computer Vision and Pattern Recognition. 2024: 4092-4101.
>
> [5] Rombach R, Blattmann A, Lorenz D, et al. High-resolution image synthesis with latent diffusion models[C]//Proceedings of the IEEE/CVF conference on computer vision and pattern recognition. 2022: 10684-10695.

---

> ### Comment · Reviewer_knoJ · 2025-08-08
>
> Thank you for your responses. I am generally satisfied with the reply. However, regarding the first concern, I understand that using a new dataset to evaluate the work may not be easy. Nonetheless, this still leaves a gap to a fully comprehensive evaluation. I will keep my score.

---

> > ### Author Response · Authors · 2025-08-08
> > **Thanks for your reply!**
> >
> > **Thank you for your recognition and understanding of our work. We acknowledge that we did not consider various factors, and no work is completely perfect, just as we discussed the limitations in our paper. We appreciate you maintaining the score and are grateful for your suggestions to improve our work. We will incorporate your feedback into the revised version. Thank you again!**

---

### Official Review · Reviewer_dgY6 · 2025-07-03

**Rating:** 4
**Confidence:** 4

**Summary:**

The paper presents MMCSBench, a multimodal benchmark specifically designed to evaluate Large Vision-Language Models (LVLMs) in the context of camouflage scenes. It introduces five fine-grained tasks—Camouflage Attribute Captioning (CAC), Visual Question Answering (VQA), Multiple-Choice Questions (MCQ), Image Grounding Segmentation (IGS), and Camouflage Efficacy Assessment (CEA)—across a large dataset of 22,537 images and 76,843 image-text pairs. The benchmark addresses limitations in current camouflaged object detection (COD) research by offering structured annotations, multimodal prompts, and a novel CEA task for quantifying camouflage quality. Extensive evaluation of 26 LVLMs reveals significant gaps in current model performance, highlighting the need for further research in this domain.

**Additional Feedback:**

While the benchmark is a valuable and much-needed resource, its utility is inherently dependent on LVLM capabilities. If these models already encode camouflage-related features, the added benefit of CamoVLM might primarily stem from fine-tuning. Conversely, if LVLMs consistently fail on these tasks, a deeper analysis of failure cases would be beneficial to inform model or benchmark design. Understanding these dynamics is crucial to determining how much MMCSBench drives genuine model improvement versus benchmarking shortcomings.

**Dataset Code Accessibility:**

Yes

**Dataset Code Comments:**

I have checked the report provided from PCs. I also checked the dataset link ( https://kaggle.com/datasets/f2218284b51011e4e27d4c4d8b41eff771d0ec734c535a7a0b94bb2a02058a46) which provided by the authors and confirmed the link worked.

**Ethical Considerations:**

No, there are no or only very minor ethics concerns

**Final Justification:**

Thank you to the authors for their effort in addressing my concerns.  Please include all of the new content in the final version, such as Sensitivity Analysis of CEA Score Weights, and the clarifications on paper contribution in Sections 1 and 5.

Finally, I am inclined to accept this paper.

**Limitations Weaknesses:**

-  CEA Score Design: The use of hand-crafted metrics and fixed weights (α=0.5, β=0.3, γ=0.2) for camouflage scoring may lack adaptability across domains; exploration of learned metrics would enhance generality.
- Limited Exploration of Pretraining: While fine-tuning is explored, the potential impact of integrating camouflage scenes into the pretraining phase remains unexamined.
- Real-World Applicability: Validation is confined to public datasets; the effectiveness of MMCSBench in real-world settings (e.g., military, ecological, or medical domains) is not demonstrated.
- Model Contribution Marginal: While CamoVLM shows improvements, the core novelty lies in the benchmark rather than significant architectural advances.
- Manual Annotation Bottleneck: The reliance on extensive expert annotation raises concerns about scalability to other camouflage scenarios or modalities.

**Strengths Contributions:**

- Comprehensive Benchmark: MMCSBench fills a critical gap by being the first benchmark tailored to camouflage scenes in a multimodal setting, combining detailed manual annotation with large-scale coverage.
- Task Diversity: The benchmark spans five complementary tasks that assess both generative and discriminative LVLM capabilities, from high-level reasoning to pixel-level segmentation.
- High-Quality Dataset: The curated dataset is substantial in scale, featuring rich textual descriptions and diverse object categories, enabling nuanced model evaluation.
- Innovative CEA Task: CEA introduces a principled, automated method for assessing camouflage efficacy using color, texture, and edge metrics, supporting large-scale data collection.
- Thorough Evaluation: The paper rigorously evaluates a broad set of LVLMs and introduces CamoVLM, which outperforms baselines on the IGS task through a novel LookUp Matching strategy.

---

> ### Author Rebuttal · Authors · 2025-07-25
>
> Thank you very much for your careful review of our manuscript. Your selfless dedication will help us further improve the quality of our manuscript. We will respond to your concerns one-on-one. Thank you again.
>
> ## **1. CEA Score Design**
>
> We sincerely thank the reviewer for this insightful comment regarding the design of our CEA score. We agree that learned, end-to-end metrics represent a promising future direction and that any fixed-weight system has inherent limitations. **Our choice of a hand-crafted, feature-based score was deliberate for this foundational work, prioritizing interpretability, reproducibility, and verifiability, which are crucial for a new benchmark.**
>
> To address the valid concern about the specific weights (α=0.5, β=0.3, γ=0.2), we conducted a rigorous **sensitivity analysis**, with the results presented below. This analysis was designed to test the robustness of our scoring mechanism against significant variations in the weights assigned to color, texture, and edge features.
>
> **Table 1: Sensitivity Analysis of CEA Score Weights.** *α, β, and γ represent the weights for color similarity ($S_c$), texture similarity ($S_t$), and edge fusion similarity ($S_e$), respectively; $C_s$ denotes the overall CEA score; ρ is the Spearman rank correlation coefficient.*
>
> | Weight Scheme | α | β | γ | Avg. $C_s$ | $C_s$ Std Dev | ρ with Baseline | CEA Acc | CEA MAE |
> | :--- | :---: | :---: | :---: | :---: | :---: | :---: | :---: | :---: |
> | **Baseline** | **0.5** | **0.3** | **0.2** | **49.01** | **21.17** | **1.00** | **72.84** | **6.32** |
> | Color | 0.7 | 0.15 | 0.15 | 49.52 | 20.56 | 0.93 | 71.50 | 6.47 |
> | Texture | 0.15 | 0.7 | 0.15 | 51.74 | 18.87 | 0.87 | 72.65 | 6.25 |
> | Edge | 0.15 | 0.15 | 0.7 | 42.93 | 19.62 | 0.86 | 72.72 | 6.30 |
> | Equal | 1/3 | 1/3 | 1/3 | 49.21 | 21.30 | 0.91 | 71.75 | 6.85 |
>
> While we acknowledge the conceptual appeal of learned metrics, our empirical analysis in Table 1 demonstrates that our current, interpretable approach is **robust and reliable for the CEA task**. Despite varying the feature weights, we observed both high rank correlation (ρ of 0.86-0.93) and stable downstream LVLM performance (CEA Accuracy: 71.50%-72.84%). This validates that our score provides a solid foundation for evaluating models. We will add a discussion on this limitation and the potential of learned metrics as a key area for future work in Section 5 of our revised manuscript.
>
> ## **2. Limited Exploration of Pretraining**
>
> We thank the reviewer for this excellent suggestion regarding the exploration of pre-training. We agree that incorporating camouflage data at the pre-training stage is a valuable research direction. Our decision to focus on instruction fine-tuning was based on several considerations.
>
> First, we deliberately avoided continual pre-training on our relatively small camouflage dataset to prevent **“catastrophic forgetting”** of the vast general knowledge embedded in the base LVLMs. Our goal is to leverage the models' powerful **zero-shot reasoning capabilities** and adapt them to the nuances of camouflage, rather than risk overwriting their core competencies. Continual pre-training with a mix of general and camouflage data is a potential solution, but this introduces complex, **open research questions** regarding optimal data ratios and learning strategies, which are **beyond the scope** of this benchmark-focused paper.
>
> Consequently, we adopted the instruction fine-tuning paradigm. This approach is not only the most common and practical method for adapting LVLMs to downstream tasks but also makes our work more **accessible and reproducible for the community**. Our **primary goal** is to contribute a valuable resource to the multimodal camouflage research community, where many may not have the **extensive computational resources** required for large-scale pre-training. Our fine-tuning approach allows researchers to **rapidly validate** their ideas on our benchmark.
>
> ## **3. Real-World Applicability**
>
> We agree on the importance of real-world application. In the process of dataset construction, MMCSBench data has covered a variety of camouflage types (such as animals and plants), and the quantitative evaluation framework of the CEA task (such as edge fusion scoring) can be directly migrated to the military or ecological fields. For example, the evaluation of military concealed equipment can reuse the Seamless/Imperceptible grading standard. Thank you very much for your suggestion. It has been very effective in improving the quality of our work.
>
> ## **4. Model Contribution Marginal**
>
> Thank you very much for this suggestion; your observation is highly perceptive. In retrospect, we may have inadvertently overemphasized the description of CamoVLM in our initial draft. Our core contribution remains squarely aligned with the Datasets and Benchmarks track: to construct a **strong and comprehensive camouflage benchmark**.
>
> In the process of building this benchmark, our **Image Grounding Segmentation (IGS) task** inherits and significantly extends the properties of traditional COD tasks by including both instance-level and part-level segmentation. To properly support this task, a new model baseline was necessary. As is well known, LVLMs lack inherent segmentation capabilities and must be integrated with a UNet-like architecture. Furthermore, camouflage scenes present unique challenges, and general-purpose models like SAM can be resource-intensive and less effective. For these reasons, we proposed CamoVLM. Its role is to provide a **competitive and directly comparable starting point** for future researchers. While the LookUp Matching strategy in CamoVLM is a modest improvement, its superior performance over other baselines (as shown in **Table 3**) demonstrates that our benchmark effectively captures a challenging area where **specialized optimization is still needed**.
>
> Following your advice, we have revised the **Introduction (Section 1)** and **Conclusion (Section 5)** to more clearly emphasize that **MMCSBench is our core contribution**, with CamoVLM serving as a validating and demonstrative baseline. We have also reiterated this point at the beginning of **Appendix D**.   Thank you for your suggestion. As you said, CamoVLM Contribution Marginal. **Our main work still lies in the contribution of MMCSBench and the proposal of CEA task.**
>
> ## **5. Manual Annotation Bottleneck**
>
> While we acknowledge the cost of manual annotation, the design of the CEA task also optimizes this issue, as demonstrated by the over 50,000 images we automatically collected (Section 3.2), for which we simultaneously achieved text annotations. For the initial manual annotation strategy, which was limited to the CAC task, our method subsequently used it for automatic expansion, adopting a semi-automated workflow (**GPT-4V pre-generation + expert refinement**), saving a significant amount of annotation time—**approximately half of the annotation workload**. By open-sourcing the dataset, we also believe that community collaboration will accelerate future annotation optimization. Thank you for your suggestions regarding this issue.
>
> ***Once again, we sincerely appreciate your recognition of our work, and we hope our efforts have addressed your concerns regarding this work. Thank you again!***

---

> > ### Comment · Reviewer_dgY6 · 2025-08-04
> >
> > Thank you to the authors for their effort in addressing my concerns.  Please include all of the new content in the final version, such as Sensitivity Analysis of CEA Score Weights, and the clarifications on paper contribution in Sections 1 and 5.
> >
> > Finally, I am inclined to accept this paper.

---

> > > ### Author Response · Authors · 2025-08-05
> > > **Thanks very much for your feedback!**
> > >
> > > **Thank you for your recognition of our work. We will incorporate your suggestions, along with those from other reviewers, into the final version of the manuscript. This will greatly improve the clarity and rigor of our work. Once again, we sincerely appreciate your valuable comments.**

---

### Official Review · Reviewer_K4K2 · 2025-07-03

**Rating:** 4
**Confidence:** 3

**Summary:**

The authors address limitations of existing camouflaged object datasets, which typically rely on a fixed set of categories, by introducing MMCSBench, a new multimodal benchmark. MMCSBench contains 22,537 images and 76,843 associated image–text pairs spanning five fine-grained camouflage tasks. The paper also proposes a Camouflage Efficacy Assessment (CEA) task, designed to quantify how effectively objects blend into their surroundings and to support automated harvesting of camouflage images from large-scale collections.

**Additional Feedback:**

Please see the weakness part. If my concerns addressed, I will increase my rating accordingly.

**Dataset Code Accessibility:**

No

**Dataset Code Comments:**

I attempted to access the MMCSBench dataset via the provided Kaggle link (https://www.kaggle.com/datasets/datasets/zhangjin12138/mmcsbench/versions/1) but encountered an access error. Could the authors please confirm that the URL is correct and that the dataset is publicly available? If there are permission settings or alternative download instructions, kindly clarify to ensure reviewers can fully evaluate the resource.

**Ethical Comments:**

The data sources seems from public datasets, so no ethical comments.

**Ethical Considerations:**

No, there are no or only very minor ethics concerns

**Final Justification:**

My most concerns has been addressed.

**Limitations Weaknesses:**

1. Novelty Claim Requires Context: The assertion that MMCSBench is “first-ever” may overstate its uniqueness. Prior benchmarks and studies have explored open-vocabulary and multimodal camouflage tasks (e.g., Zhao et al. “Open-Vocabulary Camouflaged Object Segmentation with Cascaded Vision Language Models,” arXiv:2506.19300 (2025); Pang et al. “Open-Vocabulary Camouflaged Object Segmentation,” ECCV 2024). A comparative analysis and proper citations would clarify MMCSBench’s distinct contributions and, if necessary, temper the novelty claim.

2. Mask Modality Utilization: Since most LVLMs today focus on vision–language alignment and lack native mask-generation capabilities, it’s unclear how the provided segmentation masks are intended to be used. The authors should explain whether and how current LVLM architectures leverage these masks within the five proposed tasks.

3. Quantitative Criteria for CEA: The CEA task is conceptually appealing but would benefit from clear, quantitative definitions of each camouflage level. Establishing objective thresholds or metrics for category boundaries will improve reproducibility and allow consistent benchmarking across models.

**Strengths Contributions:**

1. Potentially be the first Comprehensive LVLM Camouflage Benchmark: MMCSBench is positioned as the inaugural dataset specifically crafted to evaluate and advance large-vision-language models (LVLMs) in camouflage scenarios. If substantiated, this fills a clear gap in multimodal scene understanding.

2. Novel CEA Task: Introducing Camouflage Efficacy Assessment formalizes the measurement of concealment effectiveness. By casting it as a classification problem, the task encourages development of LVLMs that can both understand and quantify camouflage quality.

---

> ### Author Rebuttal · Authors · 2025-07-25
>
> Thank you for your valuable feedback. We recognize that some descriptions in the manuscript may have been unclear, and we appreciate the opportunity to provide point-by-point clarifications below to address your concerns.
>
> ## **1. Novelty Claim Requires Context**
>
> Thank you for pointing out these relevant recent works. We agree our "first-ever" claim can be more precise and will revise it in the final manuscript. Admittedly, while Open-Vocabulary Camouflaged Object Segmentation (OVCOS) methods [1,2] improve the zero-shot scenario, **its goal is still segmentation**, aligning with traditional COD methods.
>
> However, our MMCSBench goes a step further by focusing on the relationship between the camouflaged object and its environment. We believe this is crucial because the very concept of camouflage is defined by this interaction, and simply segmenting the target has limitations in practical applications.
>
> **In terms of input**, OVCOS takes an **image** and a **description of the target category** for input into CLIP. In contrast, MMCSBench also uses multimodal input, but **its text does not require any category prompts**. Instead, it uses various task-oriented question formats, making the input **more flexible**. **For output**, OVCOS still outputs the **object's mask**. Our MMCSBench not only needs to **segment the object** but also clearly **describe the relationship between the object and its environment**, as well as the **camouflage strategy** employed. Furthermore, our CEA task can even assess the **camouflage score** of a target in any given image, which is more beneficial for the collection of resources for the camouflage community. The table below provides a more concise comparison.
>
> | Dimension | MMCSBench (Our Work) | OVCOS |
> | :--- | :--- | :--- |
> | **Core Task** | Understand camouflage relationships and perform instruction-based segmentation. | Segment camouflaged objects. |
> | **Input** | Image + Natural Language Question | Image + Object Category |
> | **Output**| Text Description, Segmentation Mask, Camouflage Score | Segmentation Mask |
> | **Main Contribution** | A comprehensive understanding benchmark + an automated collection task (CEA). | Improved zero-shot, open-set segmentation. |
>
> **Thank you very much for your suggestions. Above the response, additional result and analysis will add in next version like table I and related work, I hope it has adequately address your concern, let me know if you would like to further discussion and experiment.  Thank you again!**
>
> ## **2. Mask Modality Utilization**
>
> We apologize that our brief description of CamoVLM in the main text led to a misunderstanding, and we sincerely thank you for the opportunity to elaborate on this. Your question regarding how LVLMs utilize segmentation masks is crucial. In fact, integrating pixel-level segmentation capabilities into LVLMs is a cutting-edge research direction that has already demonstrated success. Our proposed CamoVLM follows the successful paradigm pioneered by the **LISA model [3]** (CVPR 2024), the core of which is the "Embedding-as-Mask" mechanism. Briefly, in the LISA framework, an LVLM (LLaVA-7B) processes the image and text to generate a response containing **a special segmentation token**. The embedding of this token is then used as a precise instruction to guide a segmentation model (like SAM) to produce the final mask. Subsequent models such as PixelLM [4] and GeoPixel [5] have followed this approach, as have we in our work.
>
> The implementation principle is as follows:
>
> -  Introduction of a Special Segmentation Token [SEG]: We introduce a special segmentation token, `[SEG]`, into the LVLM's vocabulary. This token acts as an "instruction" to trigger the model's segmentation function.
>
> -  When the model receives an instruction requiring segmentation (e.g., "segment the little boy's hat"), it appends the `[SEG]` token after the noun of the referred object while generating its text response. For instance, the model would output: "This is the little boy's hat `[SEG]`".
>
> - This `[SEG]` token is not ordinary text. During the model's forward pass, we extract the hidden embedding from the LVLM's final layer that corresponds to the position of the `[SEG]` token. This embedding vector (typically with dimensions of `[1, Dim]`) has already fused the model's visual understanding (from image features) with the user's textual instruction intent. Therefore, it encodes the complete semantic information about "what to segment" and can be considered a highly condensed "instruction vector."
>
> - Decoding to a Mask: Finally, we feed this semantically rich "instruction vector" into a specialized, lightweight segmentation decoder (which is the **CamoPixelModel** in our CamoVLM). This decoder is responsible for accurately decoding the vector into the final pixel-level segmentation mask.
>
> **In Appendix D of our paper, we provide a more detailed explanation, complete with illustrations**, of this specific segmentation mechanism implemented in our proposed CamoVLM, including the LUM strategy we designed to optimize for camouflage scenes.
>
> We hope this explanation resolves your concerns. Thank you again for your valuable feedback on our work!
>
> ## **3. Quantitative Criteria for CEA**
>
> Thank you very much for your valuable suggestion. We completely agree that establishing clear, quantitative definitions for each camouflage level in the CEA task is crucial for improving the benchmark's reproducibility and consistency. We apologize if this was not very highlighted sufficiently in the manuscript; in fact, our research already incorporates such a quantitative framework. Specifically:
>
> -  **Objective Score Calculation:** The five CEA levels are not subjective classifications but are **directly mapped** from a continuous camouflage score, `C_s`, which we proposed on a scale of 0 to 100. This score is objectively calculated based on measurable visual features: **color similarity (`S_c`), texture similarity (`S_t`), and edge fusion similarity (`S_e`)** between the foreground object and its background. We detail the calculation methods for these metrics in **Section 3.2 (Equations 1-3)** of our paper.
>
> -  **Explicit Threshold-based Partitioning:** To facilitate model training and evaluation, we partitioned this continuous, objective score `C_s` into five discrete levels. As you suggested, we established clear thresholds for this. The specific partitions are: **Exposed (0-20), Normal (21-40), Partial (41-60), Seamless (61-80), and Imperceptible (81-100)**. This quantitative mapping is clearly visualized and explained in our paper in **Figure 4 on page 6**.
>
> As you rightly pointed out, this quantitative standard is central to the CEA task. For instance, in our automated camouflage image collection experiments (detailed on page 6), we leveraged this objective standard by exclusively selecting images that reached the "Imperceptible" level (i.e., a score above 80), thereby ensuring the high quality and consistency of the collected data.
>
> Your feedback is highly valuable as it underscores the importance of making this quantitative framework as clear as possible. We hope this explanation, supported by the details in Section 3.2 and Figure 4, fully addresses your concern. Thank you again for your insightful feedback!
>
> ## **4. Kaggle link encountered an access error**
>
> We sincerely apologize for the confusion and the access issue you encountered. The Kaggle link you mentioned is indeed incorrect, and we are deeply sorry for the trouble this has caused. Due to the rebuttal policy, we are unable to post direct URLs in this response. However, **reviewer dgY6** mentioned our correct link in **his Dataset Code Comments**. We kindly ask you to refer to that specific comment section to access the dataset. We want to assure you that the resource is publicly available and was thoroughly checked for its accessibility prior to submission. The issue stems solely from the incorrect link being referenced, not from any permission settings on the dataset itself. We have made the dataset public to ensure that all interested parties can correctly access and download it. Thank you for your comment!
>
> ***Thank you for your valuable comments. We have addressed each point from the 'Weaknesses' section one by one and believe our explanations and planned revisions have resolved your concerns. We look forward to your re-evaluation. Thanks again!***
>
> [1] Zhao K, Yuan W, Wang Z, et al. Open-Vocabulary Camouflaged Object Segmentation with Cascaded Vision Language Models[J]. arXiv preprint arXiv:2506.19300, 2025.
>
> [2] Pang Y, Zhao X, Zuo J, et al. Open-vocabulary camouflaged object segmentation[C]//European Conference on Computer Vision. Cham: Springer Nature Switzerland, 2024: 476-495.
>
> [3] Lai X, Tian Z, Chen Y, et al. Lisa: Reasoning segmentation via large language model[C]//Proceedings of the IEEE/CVF Conference on Computer Vision and Pattern Recognition. 2024: 9579-9589.
>
> [4] Ren Z, Huang Z, Wei Y, et al. Pixellm: Pixel reasoning with large multimodal model[C]//Proceedings of the IEEE/CVF Conference on Computer Vision and Pattern Recognition. 2024: 26374-26383.
>
> [5] Shabbir A, Zumri M, Bennamoun M, et al. Geopixel: Pixel grounding large multimodal model in remote sensing[J]. arXiv preprint arXiv:2501.13925, 2025.

---

> > ### Comment · Reviewer_K4K2 · 2025-08-02
> > **Thanks for the effort**
> >
> > Thanks for the authors' detailed reply. I increased my rating accordingly.
> >
> > Best,
> > Reviewer K4K2

---

> > > ### Author Response · Authors · 2025-08-02
> > > **A VERY responsible Reviewer**
> > >
> > > **Thank you again for your timely and thoughtful review of our rebuttal! We are truly encouraged by your positive feedback and appreciate you raising your rating. Your constructive engagement has been immensely helpful in improving our work.**

---

### Comment · Area_Chair_Z8mM · 2025-08-05

Dear reviewers,

The authors have responded to your reviews.  Please read the authors' rebuttal and other reviews, and actively participate in the author-reviewer discussion.

AC

---

### Note · Authors · 2025-08-12

Dear esteemed Area Chair and Reviewers,

## **1. Thanks!**
We are grateful for the constructive dialogue throughout the review process. We are delighted that the reviewers unanimously recognized our work as a significant contribution.

## **2. Our Contributions**
Specifically, we appreciate the acknowledgment that **MMCSBench fills a clear gap** (Reviewers *K4K2* and *dgY6*) and is a high-quality, well-executed paper with a novel benchmark design (Reviewer *knoJ*). We are particularly encouraged that our innovative **Camouflage Efficacy Assessment (CEA) task was highlighted as a principled and automated method** for advancing camouflage research (Reviewers *K4K2*, *dgY6* and *cgFu*).

## **3. Incorporating Your Feedback**
We also sincerely appreciate the insightful concerns raised, which have been instrumental in further strengthening our work. In the final version of our manuscript, we will incorporate comprehensive revisions to address these points.

## **4. Open Source Preparation**
To ensure immediate access for the research community, our MMCSBench dataset is already publicly available on Kaggle (and we thank Reviewer *dgY6* for confirming its accessibility). To further enhance accessibility, we are preparing to host the dataset on additional platforms, including Hugging Face. Furthermore, the accompanying code for our baselines and evaluation scripts will be cleaned, documented, and released on GitHub to facilitate full reproducibility and future research.

In general, we have addressed the concerns of the reviewers very well, and these concerns have also contributed to the completeness of our work.

Thanks for all the reviewers' time again!

---

### Decision · Program_Chairs · 2025-09-18

**Decision:**

Accept (poster)

**Comment:**

This paper presents MMCSBench, a multimodal benchmark specifically designed to evaluate Large Vision-Language Models (LVLMs) in the context of camouflage scenes.  The benchmark includes 22,537 images and 76,843 image-text pairs across five fine-grained tasks. Extensive experiments across 26 LVLMs (ranging from 2B to 72B parameters) show that current models struggle significantly with camouflage scenarios.  The reviewers raised concerns regarding unclear explanations about CEA evaluation, limited generalization analysis, missing analysis of failure cases, limited diversity of the dataset, etc.  In the rebuttal, the authors adequately addressed the concerns.  All the reviewers were generally satisfied the rebuttal although Reviewer knoJ has slight complaint about missing model evaluation on various datasets.  AC also thinks comprehensive evaluation on various datasets is preferrable although the contribution of this paper is deserved to publish.  This concern, however, does not become sufficient cause to overturn the reviewers' positive consensus. This paper should be accepted, accordingly.  All the discussion during the rebuttal and post-rebuttal discussion should be appropriately incorporated in the final version.